

# Observing glacier elevation changes from spaceborne optical and radar sensors – an inter-comparison experiment using ASTER and TanDEM-X data

Livia Piermattei[1,2], Michael Zemp[3], Christian Sommer[4], Fanny Brun[5], Matthias H. Braun[4], Liss M. Andreassen[6], Joaquín M.C. Belart[7], Etienne Berthier[8], Atanu Bhattacharya[9], Laura Boehm Vock[10], Tobias Bolch[11,12], Amaury Dehecq[5], Inés Dussaillant[3], Daniel Falaschi[12,13], Caitlyn Florentine[14], Dana Floricioiu[15], Christian Ginzler[1], Gregoire Guillet[12], Romain Hugonnet[1,16,17], Matthias Huss[1,17,18], Andreas Kääb[2], Owen King[19], Christoph Klug[20], Friedrich Knuth[16], Lukas Krieger[15], Jeff La Frenierre[21], Robert McNabb[22], Christopher McNeil[23], Rainer Prinz[24], Louis Sass[23], Thorsten Seehaus[4], David Shean[16], Désirée Treichler[2], Anja Wendt[25], Ruitang Yang[2]

[1]Swiss Federal Institute for Forest, Snow and Landscape Research WSL, Birmensdorf, 8903, Switzerland
[2]Department of Geosciences, University of Oslo, Oslo, 0371, Norway
[3]Department of Geography, University of Zurich, Zurich, 8057, Switzerland
[4]Institut für Geographie, Friedrich-Alexander-Universität Erlangen-Nürnberg, 91058, Germany
[5]Univ. Grenoble Alpes, IRD, CNRS, INRAE, Grenoble INP, IGE, 38000 Grenoble, France
[6]Section for Glaciers, Ice and Snow, the Norwegian Water Resources and Energy Directorate (NVE), Oslo, 0371, Norway
[7]National Land Survey of Iceland, Akranes, 300, Iceland
[8]LEGOS, Université de Toulouse, CNES, CNRS, IRD, UPS, Toulouse, 31000, France
[9]Department of Earth Sciences & Remote Sensing, JIS University, Kolkata, 700109, India
[10]St. Olaf College, Northfield, Minnesota, 55057, USA
[11]Institute of Geodesy, Graz University of Technology, Graz, 8010, Austria
[12]Department of Geography and Sustainable Development, University of St.Andrews, KY16 9AL, UK
[13]Instituto Argentino de Nivología, Glaciología y Ciencias Ambientales, CCT-Mendoza CONICET, 5500, Argentina
[14]U.S. Geological Survey, Northern Rocky Mountain Science Center, Bozeman, Montana, 59715, USA
[15]Remote Sensing Technology Institute, German Aerospace Center (DLR), Oberpfaffenhofen, 82234, Germany
[16]University of Washington, Civil and Environmental Engineering, Seattle, WA, 98195-2700, USA
[17]Laboratory of Hydraulics, Hydrology and Glaciology (VAW), ETH Zürich, Zürich, 8057, Switzerland
[18]Department of Geosciences, University of Fribourg, Fribourg, 1700, Switzerland
[19]School of Geography, Politics and Sociology, Newcastle University, NE1 7RU, UK
[20]Institute of Geography, University of Innsbruck, 6020, Austria
[21]Gustavus Adolphus College, St. Peter, Minnesota, 56082-1498, USA
[22]School of Geography and Environmental Sciences, Ulster University, Coleraine, BT15 1AP, UK
[23]U.S. Geological Survey, Alaska Science Center, Anchorage, AK, 99501, USA
[24]Department of Atmospheric and Cryospheric Sciences, University of Innsbruck, 6020, Austria
[25]Bavarian Academy of Sciences and Humanities, Munich, 80539, Germany

*Correspondence to*: Livia Piermattei (livia.piermattei@wsl.ch)

**Abstract.** Observations of glacier mass changes are key to understanding the response of glaciers to climate change and related impacts, such as regional runoff, ecosystem changes, and global sea-level rise. Spaceborne optical and radar sensors make it possible to quantify glacier elevation changes, and thus multi-annual mass changes, on a regional and global scale. However,



estimates from a growing number of studies show a wide range of results with differences often beyond uncertainty bounds. Here, we present the outcome of a community-based inter-comparison experiment using spaceborne optical stereo (ASTER) and synthetic aperture radar interferometry (TanDEM-X) data to estimate elevation changes for defined glaciers and target periods that pose different assessment challenges. Using provided or self-processed digital elevation models (DEMs) for five test sites, 12 research groups provided a total of 97 spaceborne elevation-change datasets using various processing strategies. Validation with airborne data showed that using an ensemble estimate is promising to reduce random errors from different instruments and processing methods, but still requires a more comprehensive investigation and correction of systematic errors. We found that scene selection, DEM processing, and co-registration have the biggest impact on the results. Other processing steps, such as treating spatial data voids, differences in survey periods, or radar penetration, can still be important for individual cases. Future research should focus on testing different implementations of individual processing steps (e.g. co-registration) and addressing issues related to temporal corrections, radar penetration, glacier area changes, and density conversion. Finally, there is a clear need for our community to develop best practices, use open, reproducible software, and assess overall uncertainty in order to enhance inter-comparison and empower physical process insights across glacier elevation-change studies.

## 1 Introduction

The geodetic mass balance of a glacier is derived by first calculating its volumetric change through time from repeated topographic surveys of surface elevation and surface extent and subsequently multiplying by the firn and ice density (Cogley et al., 2011). Various methods are available to assess the geodetic changes of a glacier, including traditional mapping techniques and topographic maps (e.g., Reinhardt and Rentsch, 1986; Joerg and Zemp, 2014). The most commonly used approach is to calculate the elevation difference between digital elevation models (DEMs) created for the entire glacier area, which enables the measurement of elevation and volume changes of the glacier surface over time (Cogley et al., 2011). Depending on the spatial scale, multi-temporal DEMs for geodetic assessments have been generated from terrestrial, airborne, and spaceborne platforms using optical, lidar (Light Detection and Ranging), and radar (Radio Detection and Ranging) sensors. Compared to terrestrial and airborne surveying, spaceborne sensors have opened the possibility of calculating elevation changes not only on individual glaciers but also on glacier samples covering entire mountain ranges or glacierized regions. The reader is referred to Berthier et al. (2023, and references therein) for a review of spaceborne sensors, processing techniques, and applications. Generally, optical and radar DEMs have been the two (Abrams, 2000) main sources to observe elevation changes at local, glacier, regional, and global scales. Optical stereo DEMs with operational global coverage are mainly derived from ASTER (Advanced Spaceborne Thermal Emission and Reflection Radiometer; Abrams 2000) images, acquired from 2000 onwards. Early studies have demonstrated the potential of ASTER for assessing glaciological change (e.g., Kääb, 2002). Since 2016, the free availability of ASTER images, together with the development of open-source automated processing chains like MicMac ASTER (MMASTER; Girod et al., 2017) incorporating the ASTER model and Ames Stereo Pipeline (ASP;



Beyer et al., 2018), has boosted their use within the scientific community. Consequently, glacier elevation-change assessments from ASTER have become available for large regions (Brun et al., 2017; Dussaillant et al., 2019; Shean et al., 2020), and for global coverage (Hugonnet et al., 2021).

The production of (almost) global DEMs from spaceborne bistatic interferometric synthetic aperture radar (InSAR) imagery began with the Shuttle Radar Topography Mission (SRTM; Farr et al., 2007). This experimental mission took place between the 11th and 22nd of February 2000, utilising C-Band technology to reconstruct a continuous moderate-resolution DEM for a latitude range from 56° south to 60° north. A decade later, the TanDEM-X satellite constellation emerged using bistatic X-band data (Wessel et al., 2018), offering high-precision, timely, and globally consistent DEMs. By processing individual bistatic TanDEM-X acquisitions, timestamped DEMs can be generated, enabling the creation of multi-temporal assessments of glacier elevation changes at decadal or multi-year intervals (Abdel Jaber et al., 2019; Braun et al., 2019).

With ASTER and TanDEM-X, the scientific community now has both optical and radar missions in space that enable the assessment of all glacier elevation changes worldwide over the past two decades. However, both systems come with limitations and methodological differences. As such, shadow, snow, polar night, saturation over bright surfaces and cloud cover challenge the use of optical images from ASTER, which has only an 8-bit radiometric resolution. These limitations reduce the temporal coverage and the quality of the resulting DEMs (e.g., Hugonnet et al., 2021). Similarly, the use of InSAR data is challenged by radar layover effects (Kropatsch and Strobl, 1990) and phase unwrapping errors in steep terrain (Dehecq et al., 2016; Lachaise et al., 2012), as well as radar signal penetration on glaciers (Rignot et al., 2001; Berthier et al., 2006). Several studies have conducted regional comparisons between ASTER and InSAR results and have observed significant differences in glacier elevation change rates (e.g., Dussaillant et al., 2018; Hugonnet et al., 2021; IPCC, 2021). Direct comparison of results from both systems is not only hampered by methodological differences, but can also be complicated by other factors like spatial and temporal coverage of the satellite imagery, data samples, and differences in the applied processing chains. This motivates formal inter-comparison experiments, designed to parse and quantify the impacts of data selection and processing techniques.

In this study, we present the results from an inter-comparison experiment on glacier elevation changes, which was organised within the framework of the Regional Assessments of Glacier Mass Change (RAGMAC) working group of the International Association of Cryospheric Science (IACS, 2023). International research groups were invited to participate in an open call to compute glacier elevation changes and related uncertainties for defined glaciers and time periods using a provided sample of ASTER, TanDEM-X, and SRTM DEMs. The overall aim was to assess how consistently and accurately glacier elevation changes, and thus volume and mass changes, can be estimated from spaceborne optical and radar data. Therefore, the first experiment aimed to compare the satellite-based estimates of glacier elevation change provided by the participants against airborne validation data for individual glaciers over a consistent target period. In a second experiment, we aimed to understand better the impact of different processing steps such as bias corrections, co-registration, outlier filtering, void filling, radar signal penetration, and temporal corrections on the elevation change estimates for selected large glaciers. We describe the setup of this community-based inter-comparison experiment, providing an extensive description of the DEM processing strategies adopted by the participants to derive their spaceborne results. We summarise the lessons learned from the experiments and



discuss the need for future research to improve consensus estimates of regional and global geodetic glacier mass balance from spaceborne optical and radar sensors.

## 2 Data

We describe below the study sites, the airborne validation data, ASTER, TanDEM-X, and SRTM DEMs, as well as auxiliary data provided for the experiments, such as the glacier outlines, Copernicus DEMs used as reference elevation data and glaciological time series for temporal corrections. For the present study, all topographic data were transformed to the WGS84 datum and UTM projections using bilinear interpolation.

### 2.1 Study site selection

The inter-comparison experiments are conducted on five study sites (Fig. 1) chosen based on their different characteristics such as glacier size, topography, location, and availability of validation data. Furthermore, the selected sites pose various data processing challenges for both optical and radar sensors. The individual glaciers with airborne validation data are Hintereisferner (Austrian Alps), Grosser Aletschgletscher (Swiss Alps), and Vestre Svartisen (Vestisen) Ice cap (Scandes, Norway), hereafter referred to as Hintereis, Aletsch, and Vestisen, respectively. In order to include larger experiment sites, we

also selected the Baltoro Glacier (Karakoram, Pakistan), here named Baltoro, and the Northern Patagonian Icefield (Andes, Chile).

Hintereis, located in the Ötztal Alps, and Aletsch, the largest glacier in the European Alps, are extensively studied glaciers that exemplify typical Alpine valley glaciers, which have displayed fast retreats and consistent mass loss over the last decades. Challenges for DEM differencing in the Alpine sites are the relatively small glacier size of Hintereis, with a length of

125 approximately 7 km and a glacier tongue width of less than 600 m, the relatively large and high accumulation area of Aletsch, which makes it prone to radar penetration, and the steep slopes of the surrounding terrain, which characterise both glaciers. Vestisen, an ice cap located near the polar circle in Norway, is frequently covered by clouds limiting the observation from optical satellites, and the presence of limited stable terrain surrounding the glacier adds challenges to the DEM production and post-processing steps. The accumulation area poses specific challenges for both optical satellite photogrammetry due to

130 textureless surface and winter radar acquisition, which are prone to surface radar penetration.

Baltoro, the second-largest glacier in South West Asia (Windnagel et al., 2023), was selected due to its large size, which requires the mosaicking of several DEMs. Furthermore, high-relief topography can cast shadows in the optical images and cause geometric DEM distortions in radar images. Winter radar acquisitions are also affected by radar penetration, which can bias the elevation change signal on the glacier surface. The Northern Patagonian Icefield, located in the Southern Hemisphere,

is the largest study site composed of dozens of glaciers. It is of particular interest for the use of SRTM, as surface melting conditions are expected for February 2000 (Jaber et al., 2013) and thus radar signal penetration is assumed to be negligible (Braun et al., 2019; Dussaillant et al., 2018). In addition to its size, the second-largest glacier complex in the Southern Andes



(Windnagel et al., 2023) presents all the challenges of the other study sites for both optical and radar DEM differencing techniques. These include steep and rough adjoining terrain, a smooth and extensive accumulation zone, a textureless central plain icefield, frequent cloud cover on the western side due to the maritime climate and clearer conditions on the eastern outlet glaciers.

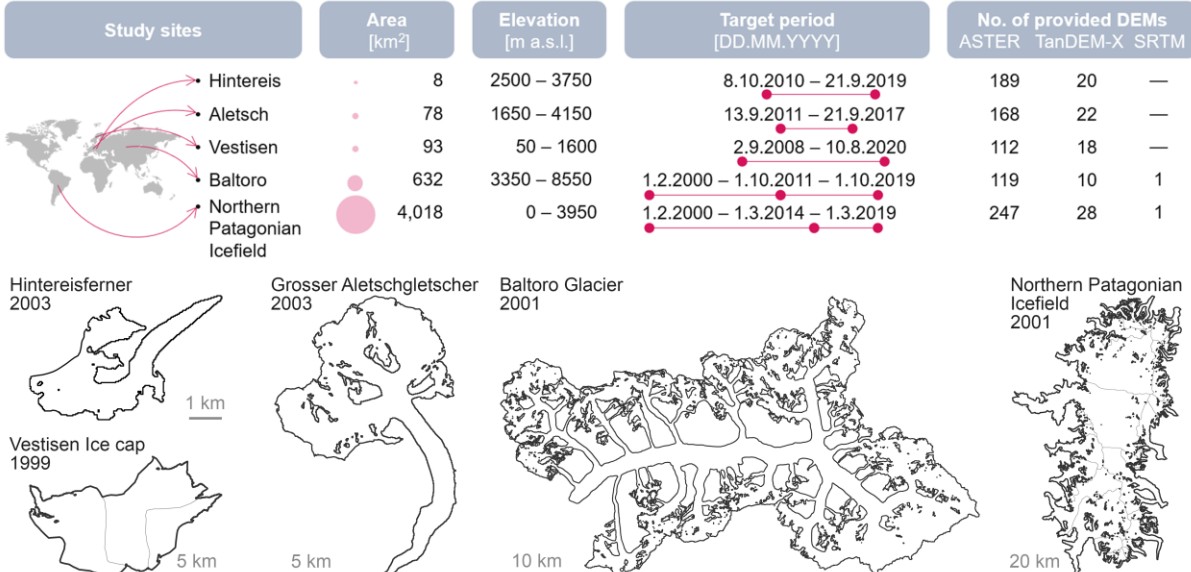

**Figure 1: Illustration of the location, area, and elevation range of the investigated glacier sites together with the corresponding target periods and the number of provided ASTER, TanDEM-X, and SRTM DEMs. Glacier outlines from RGI version 6.0, with corresponding survey years, are shown for each site.**

## 2.2 Glacier outlines

For comparison purposes, all analyses, including validation data estimates, were performed using glacier outlines from the Randolph Glacier Inventory (RGI) version 6.0 (RGI Consortium, 2017). Therefore, elevation-change estimates were derived over the same fixed glacier area. Note that the RGI outlines (Fig. 1) typically date from before the target periods and, hence, do not capture glacier area changes over time. Consequently, the calculated specific elevation changes are potentially biased in regions with large area changes, i.e. in the Alps. Original sources of the glacier outline assimilated into the RGI database for the study glaciers are Paul et al. (2011) for Hintereis and Aletsch, Andreassen et al. (2012) for Vestisen, Mölg et al. (2018) for Baltoro and Rivera et al. (2007) for Northern Patagonian Icefield.

## 2.3 Airborne validation DEMs

Validation data from airborne lidar and aerial stereo images are available for Hintereis, Aletsch, and Vestisen. These data were collected at a specific time towards the end of the melt season, which defines the target assessment periods during which spaceborne data should match. Compared to spaceborne observations, the selected airborne DEMs have high spatial resolution and quality, with minimal noise and data voids within the observed area. Moreover, they are expected to offer higher accuracy



and to not be subject to signal penetration. However, it is important to note that they do present certain issues specific to each
study site.

Airborne lidar data for Hintereis were acquired on October 8 2010, and September 21 2019 (Table S1). The 2010 lidar DEM
was provided with a resolution of 1 m (Bollmann et al., 2015), whereas the 2019 DEM was originally provided at a higher
resolution of 0.2 m by 3D RealityMaps. After resampling the DEM to a common resolution of 1 m, a co-registration between
the two DEMs was performed on stable terrain to minimize systematic offsets (Fig. S1). The least squares matching algorithm
implemented in the OPALS tool (Pfeifer et al., 2014) was used (Table S1).

The Aletsch DEMs were obtained from aerial stereo imagery and are dated 13 September 2011 and 21 September 2017 (Table
S2), although in both cases the aerial survey periods spread over one month and different time-stamped DEMs had to be
combined to cover the entire glacier (Fig. S2). The 2011 DEM was generated by the Swiss Federal Institute for Forest, Snow
and Landscape Research (WSL) at a resolution of 1 m based on images with a ground sample distance of 0.5 m (Ginzler and
Hobi, 2015), obtained from the Swiss Federal Office of Topography (SwissTopo). The 2017 DEM was provided at a resolution
of 2 m and was directly downloaded from SwissTopo (2023). Despite the different acquisition dates over the glacier, no
artefacts or jumps are detectable in the DEMs, as shown in the longitudinal profile in Fig. S3. No co-registration was applied
because of the symmetric distribution of elevation changes on stable terrain, with a mean value of -0.03 m and NMAD of less
than 1 m, and not visible shift on the elevation change map (Fig. S4).

On Vestisen, airborne lidar data was collected with a point density of more than two points per square metre for surveys on 2
September 2008 and 10 August 2020 by the companies Blom Geomatics and TerraTec, respectively (Table S3). DEMs were
generated at a resolution of 10 m and are available from the Norwegian mapping authorities. A co-registration on stable off-
glacier terrain was carried out following the same approach as used for Hintereis (Fig. S5).

For the three sites, random errors in the mean glacier-wide elevation changes of the validation data were estimated considering
the spatial correlation of elevation change errors at multiple ranges and applying error propagation for correlated variables
(Hugonnet et al., 2022). Due to the high accuracy of the input DEMs, the standard deviation of elevation change over the stable
terrain is relatively small for mountain terrain (i.e., around one metre), and the resulting uncertainty at a 95 % confidence
interval of the mean elevation change over the glacier area (considering spatial auto-correlation) are ±0.18, ±0.26 and ±0.92
m for Vestisen, Hintereis and Aletsch, respectively. A more detailed description of the validation data, post-processing steps,
and uncertainty estimates are reported in supplementary Tables S1, S2, and S3. For Baltoro and Northern Patagonian Icefield,
no validation data was available.

## 2.4 Copernicus reference DEM

The Copernicus DEM (ESA and Airbus, 2022) is provided for individual glaciers as a reference DEM for co-registration and
filtering, but not for estimating glacier elevation change. It was chosen as reference DEM because it is openly available with
global coverage at 30 m resolution, and without data voids. The Copernicus DEM is mainly based on SAR interferometry from



the TanDEM-X radar satellite data acquired between December 2010 and January 2015. For each study site, we cropped the global Copernicus DEM including sufficient stable (i.e., not-glacierized) terrain for co-registration.

## 2.5 Spaceborne experiment DEMs

### 2.5.1 ASTER DEMs

The Terra satellite (EOS AM-1) was launched in December 1999 on a sun-synchronous orbit and is currently in a constellation exit with expected to decommission in ca. 2025, without being replaced. It carries several sensors including the ASTER system, which collects pairs of stereo images globally at a spatial resolution of 15 m in the near-infrared band, making its data the largest consistent multi-temporal dataset of stereo images available worldwide (Girod et al., 2017). The stereo pairs consist of a nadir-pointing image (Band 3N) and a backward-looking image (Band 3B) with an effective parallax angle of 30.6°. The

average elevation precision of ASTER DEMs is reported to be around ±5−10 m on flat terrain and ±15−20 m on steeper terrain (Kääb, 2002; Hirano et al., 2003; Toutin, 2008).

For each study site, we downloaded all the ASTER L1A images with less than 80 % cloud coverage from the National Aeronautics and Space Administration (NASA) Land Processes Distributed Active Archive Center via EarthData Search (ASTER L1A Reconstructed Unprocessed Instrument Data, version 003. DOI: 10.5067/ASTER/AST_L1A.003, 2023). The

images have a tile size of about 60 km by 60 km. The DEMs were generated using the Ames Stereo Pipeline (ASP, Beyer et al., 2018). We used a block-matching algorithm with a correlation kernel of seven pixels and a kernel value of 13 for the subpixel refinement. The algorithm is run on the images projected onto the Copernicus DEM (Sect. 2.4) of the area to avoid large-scale distortions.

For the selected sites, ASTER DEMs were provided to the participants at a resolution of 30 m, without any filtering, and data

voids were not filled or interpolated. ASTER data have been available since 2000, and DEMs are generated from that year onwards. The number of ASTER DEMs available for the sites is high, with at least one DEM covering parts of the target glacier(s) per year. Coverage varies from site to site, ranging from 112 (Vestisen) to 247 (Northern Patagonian Icefield) DEMs. The main challenge when using ASTER DEMs is the incomplete spatial coverage on the glacier for the same date. This is due to cloud cover, saturation of the 8-bit sensor in snow-covered areas, and the acquisition footprint of the instrument. The use of

a block-matching correlation algorithm with a small window leads to DEMs with large voids in the accumulation area (especially for Vestisen and Northern Patagonian Icefield). For example, in the case of Hintereis, for a total of 189 provided DEMs, the number of available DEM data points varies from 26 to 73 per glacier pixel (Fig. 2a). The spatio-temporal coverage of the experiment's DEMs over the other study sites is shown in the appendix material (Figs. A1 to A4).





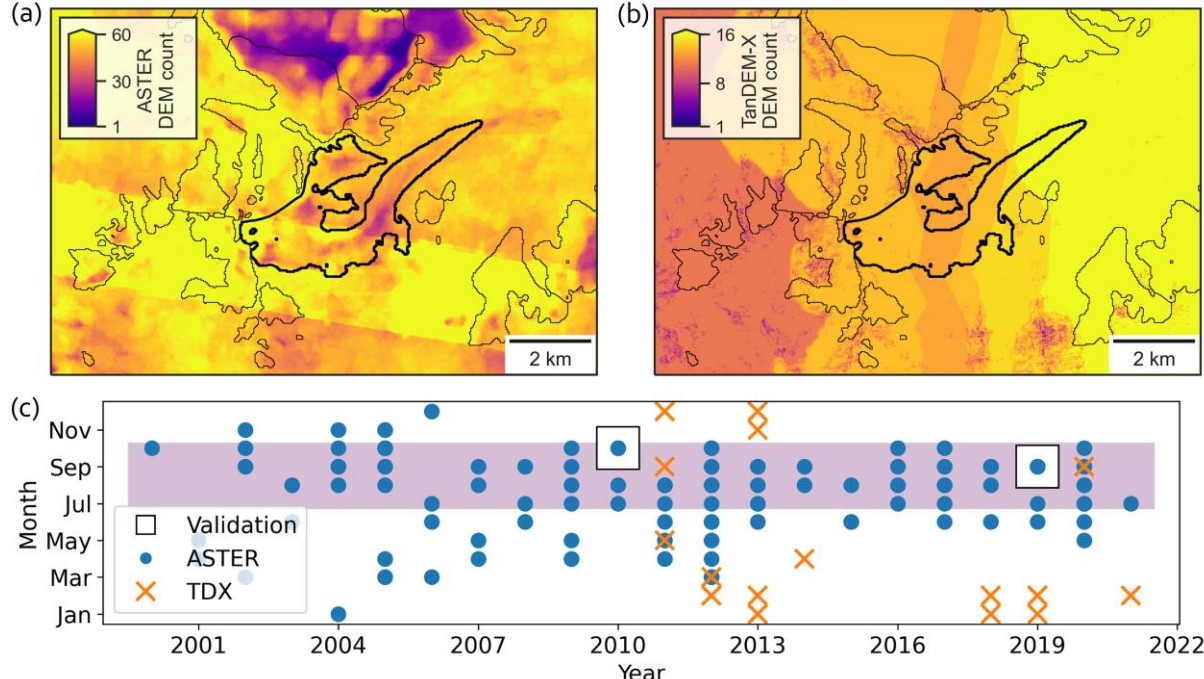

**Figure 2: Spatio-temporal coverage of the experiment DEMs over Hintereis (bold RGI 6.0 outline). The DEM count over the area is shown for a) ASTER and b) TanDEM-X (TDX). c) Dates of the validation period and temporal coverage of the TanDEM-X and ASTER DEMs provided for Hintereis. The summer months (July to October) are highlighted in purple.**

### 2.5.2 TanDEM-X DEMs

InSAR DEMs are retrieved from the TerraSAR-X add-on for the Digital Elevation Measurement mission (TanDEM-X) (Krieger et al., 2007). The mission is jointly operated by the German Aerospace Center (DLR) and Airbus Defence and Space. Interferometric X-band SAR data have been acquired since 2010 with two global DEM coverages and additional campaign-based acquisitions (TanDEM-X - A New High Resolution Interferometric SAR Mission, 2023b). For DEMs created from TanDEM-X images, elevation precision is reported to be better than the specified ±10 m (Wessel et al., 2018) but is expected to be less accurate in mountainous terrain (Rizzoli et al., 2017).

A total of ~220 TanDEM-X Coregistered Single look Slant range Complex acquisitions between December 2010 and February 2021 were downloaded from the DLR Earth Observation Center (EOWEB GeoPortal, 2023a) and processed following the interferometric workflow described by (Braun et al., 2019).

Initially, overlapping TanDEM-X scenes of the same acquisition path and date are concatenated into continuous DEM strips. Differential interferograms are created from each TanDEM-X acquisition strip using the GAMMA remote sensing software environment (Werner et al., 2000). After that, each interferogram is filtered and unwrapped based on the minimum-cost-flow algorithm of the GAMMA Interferometric SAR processor module and converted into elevation values by adding respective reference surface heights. The reference surface elevations of the Alpine test sites, Northern Patagonian Icefield, and Baltoro



glacier are extracted from the void-filled SRTM DEM (Sect. 2.5.3), while for Vestisen we used the Copernicus DEM (Sect. 2.4). Each TanDEM-X DEM strip was visually inspected and acquisitions with large-scale distortions or errors (e.g. phase

jumps) were removed from the final selection. Finally, the selected DEMs were resampled to 10 m spatial resolution. No further post-processing steps were applied to the selected DEMs.

TanDEM-X DEMs provide almost complete coverage on the glacier (~99%; Fig. 2, A1−A4), but there are voids mainly off-glacier due to radar shadows or layover that were masked out during the DEM production. It was not possible to provide annual coverage of TanDEM-X DEMs in all cases due to the campaign-based acquisition mode of the mission besides the multi-

annual global DEM acquisition efforts. The various acquisitions are also subject to different interferometric baselines which results in a variable height sensitivity. Note that the acquisition months vary between study sites due to the acquisition strategy of the mission.

### 2.5.3 SRTM DEMs

The SRTM of NASA provides a seamless DEM of all land masses outside the polar regions with a vertical accuracy of 16 m

(Farr et al., 2007). The SRTM is used as reference surface elevation for TanDEM-X production and as an InSAR DEM for estimating the elevation change of the Baltoro and Northern Patagonian Icefield for the period 2000−2010s. We use the void-filled LP DAAC NASA V.3 SRTM DEM provided by USGS Earth Resources Observation and Science (EROS) with a spatial resolution of 30 m (Podest and Crow, 2013). In the void-filled version, data voids due to layover and shadow were filled with existing terrain heights from different sources (USGS, 2017).

### 2.6 Glaciological data for temporal corrections

In-situ surface mass-balance measurements provided by the World Glacier Monitoring Service (WGMS, 2021) were used by some groups for the annual correction of the geodetic estimates from the satellite DEMs to approximately match the target period defined by the airborne validation data. Among the selected glaciers, glaciological mass-balance measurements are available for Aletsch (Huss et al., 2015; GLAMOS, 2022), Hintereis (Klug et al., 2018), and Engabreen (Kjøllmoen et al.,

2021, and earlier reports), which is one of the outlet glaciers of the Vestisen study site. These glaciological observations were also used for the temporal corrections (Sect. 3.3.6) applied to those spaceborne results of Hintereis, Aletsch and Vestisen where the reported experiment results did not match the exact target periods. The mass-balance time series used in the temporal corrections were calibrated to the airborne geodetic validation results following the approach by (Zemp et al., 2013). For density conversion, we used $850 \text{ kg m}^{-3}$ following Huss (2013), noting that for short time periods, particularly during winter

snow accumulation, this assumption comes with large uncertainties.



## 3 Methods

### 3.1 Inter-comparison experiment description

DEMs from ASTER and TanDEM-X were generated (Fig. 3, step 1) for each study site and provided to the participants, together with auxiliary data, excluding airborne validation data. SRTM DEM was only provided for Baltoro and Northern
Patagonian Icefield. Within the inter-comparison experiment, participants applied different DEM selection strategies (Sect. 3.3.1), post-processing steps, and corrections (Fig. 3, steps 2 to 4) to calculate their spaceborne estimates.

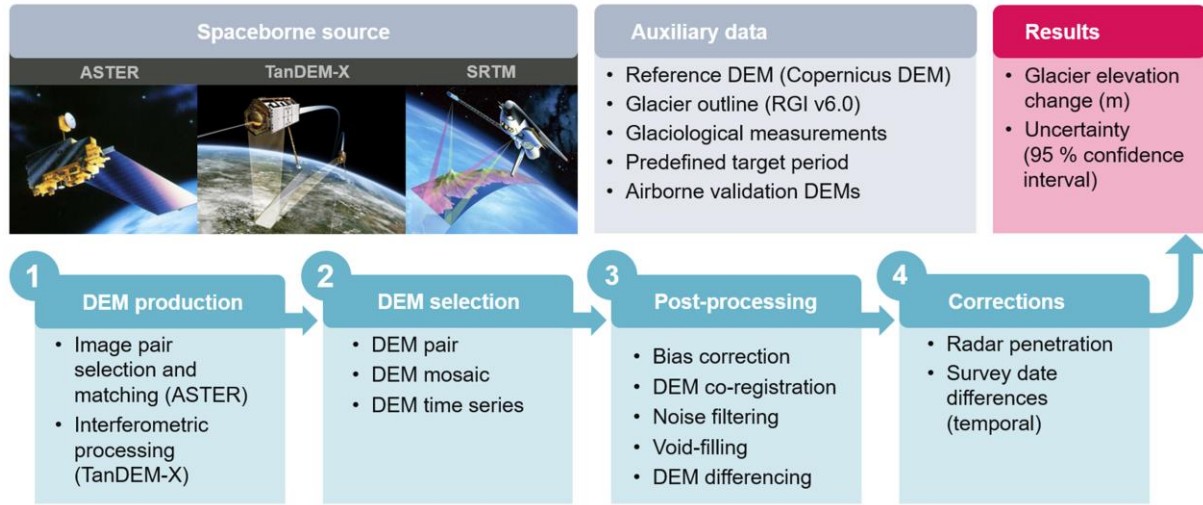

**Figure 3: Experiment configuration and general workflow for glacier elevation change assessment using DEM differencing from optical (ASTER) and radar (TanDEM-X and SRTM) spaceborne data. Grey boxes summarise spaceborne and auxiliary data, light**
**blue boxes summarise DEM processing steps, and the red box summarises results for the inter-comparison experiments. ASTER image source: NASA JPL; TanDEM-X image source: DLR; SRTM image source: Detlev Van Ravenswaay.**

The spaceborne results submitted by the participants are the mean glacier-wide elevation changes and related uncertainties for selected study sites using all or a selection of optical and/or radar DEMs for a predefined period. The inter-comparison study consists of two experiments. The first experiment aimed to quantify how well glacier elevation change can be estimated from
ASTER and TanDEM-X spaceborne data separately, in comparison to airborne validation data for a target period (Fig. 1). Thereby, the focus of the first experiment was to derive spaceborne glacier elevation-change estimates as close as possible to the period imposed by the airborne validation data, including a temporal correction if needed. This experiment was run for the three smaller sites of Hintereis, Aletsch and Vestisen. A second experiment, performed on the Baltoro and Northern Patagonian Icefield, aimed to evaluate the impact of the processing steps (i.e., steps 3 and 4 in Fig. 3) on glacier elevation change estimates
over two target periods of approximately 10 years (Fig. 1). In this sensitivity study, participants were requested to estimate the mean elevation change using their own workflow, which we will refer to as the 'reference' result, from ASTER, TanDEM-X, and SRTM DEMs. DEM from different sources could be used for this second experiment. Subsequently, the participants calculated separate mean elevation change estimates by selectively switching on and off each post-processing and correction





step. To quantify the specific impact of each step, we subtracted the estimate obtained without that step from the reference
estimate.

## 3.2 Participants and spaceborne results

The inter-comparison experiment involved 12 groups from different institutions. Supplementary Tables S4 to S15 provide a complete list of the members within each group. Figure 4 shows the three-letter acronym of the group's main institutions together with the number of spaceborne results submitted for the selected study sites.

Some groups employed different DEM sources and approaches for the same study site and thus submitted multiple results. Results are listed per unique source and/or workflow and kept as discrete submissions, rather than grouped by institution so that each and every submitted result is used in the inter-comparison study. Some results followed the experimental design but were flagged by the participants as "low confidence", following IPCC terminology (Mastrandrea et al., 2010). The criteria for such flagged results are reported in the supplementary tables. All submitted results were considered for comparison, but low-
confidence results are marked in figures and tables, and excluded from ensemble estimations.

In total, the participants contributed 97 spaceborne results. Vestisen and the Northern Patagonian Icefield received fewer submissions than the other sites (Fig. 4), likely because these sites were designated as "optional" within the experiment. It is notable that ASTER DEMs were used more often than TanDEM-X (Fig. 4), and only a few results are based on both optical and radar DEMs. Hintereis and Baltoro received the most spaceborne results and, hence, are used to illustrate the experiments
in the main body of this manuscript, with figures of the other study sites available in the supplement. We note that the ASTER results by ETH are from the global study by (Hugonnet et al., 2021), extracted at the closest month to the target periods.

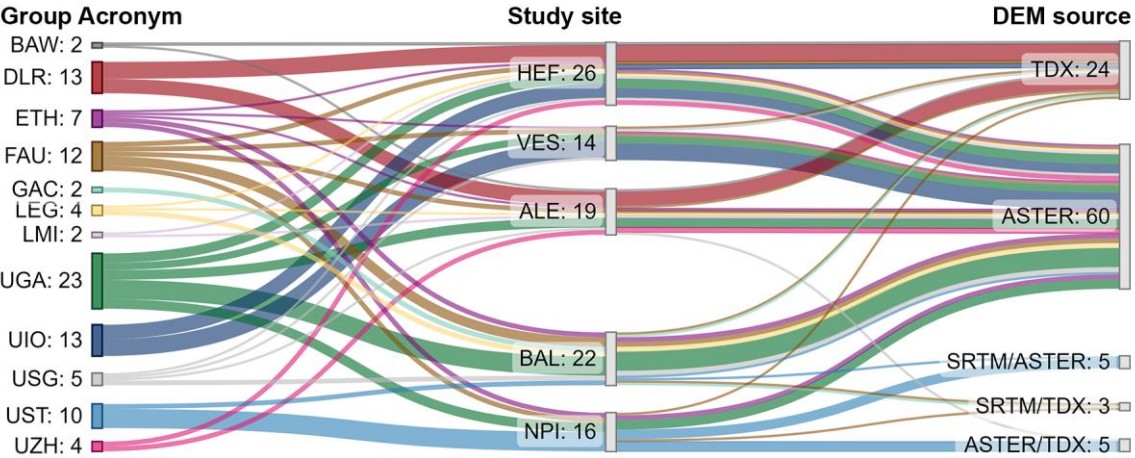

**Figure 4: Sankey diagram linking the number of results of each participating group (left), sorted alphabetically by their acronym, with the study sites (centre) − Hintereis (HEF), Vestisen (VES), Aletsch (ALE), Baltoro (BAL), and Northern Patagonian Icefield**
**(NPI) − and the DEM source (right), i.e., TanDEM-X (TDX), ASTER, SRTM, and combinations thereof. Note that for Baltoro and Northern Patagonian Icefield, the diagram includes the reference results (excluding the sensitivity results) and counts for both target periods (i.e., 2000−2010s, 2010s−2019). Sankey produced with Sankeymatic (SankeyMatic, 2023).**





### 3.3 General workflow for glacier elevation change assessment

The overall workflow to quantify the glacier elevation change using multi-temporal DEMs from spaceborne observations
consists of the four steps of DEM production, DEM selection, post-processing, and further corrections (Fig. 3). Post-processing
includes bias correction, co-registration, noise filtering, and void-filling. Further corrections can be applied such as temporal
corrections (e.g., seasonal, annual, or trend) to match the target period, or corrections for radar signal penetration. The different
steps and related implementations by the groups are explained below, while more detailed descriptions of the DEM selection
and processing strategies used by each group for each site are provided in Tables S4 to S15.

**3.3.1 DEM production, selection, and differencing strategy**

The ASTER and TanDEM-X DEMs were produced according to the methods described in Sect. 2.4.1 and 2.4.2, respectively.
Nonetheless, participants had the option of generating their own ASTER and TanDEM-X DEMs, hereafter named "self-
processed". Two groups (ETH and UIO) utilised the L1A images and the MMASTER processing chain (Girod et al., 2017) to
generate their own ASTER DEMs. The DLR group followed the workflow described by Fritz et al. (2011) to generate their
own TanDEM-X DEM. Including these results allowed us to cover the additional spread in results from the DEM processing
step.

To estimate the glacier elevation changes for the target period, three main approaches were employed based on different subsets
of multi-temporal DEMs (Fig. 5).

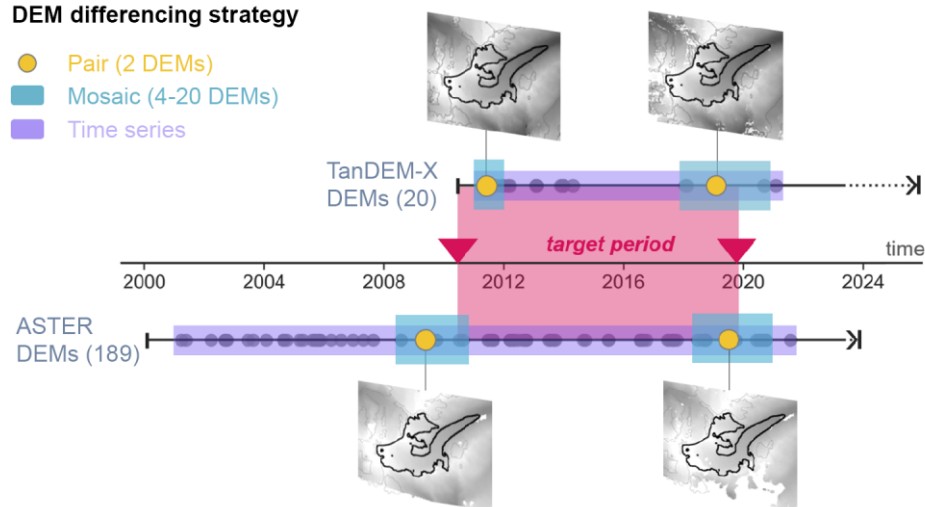

**Figure 5: Graphical illustration of DEM selection strategies using a DEM pair, a DEM mosaic, or a DEM time series to estimate
glacier elevation changes. The number of DEMs, the predefined target (validation) period, and the thumbnail of selected, provided
TanDEM-X and ASTER DEMs are shown for the Hintereis study site. Note that the selected ASTER DEM pairs were not always
the closest to the target period due to data voids.**





One approach referred to as the "DEM pair" consists of calculating the elevation difference using two single DEMs that are
close to the validation date and provide the best glacier coverage. A similar approach named "DEM mosaic" groups the DEMs
in a time window of up to a few years around each validation date. Then for each DEM group, a mean or median of the
elevation, or a mosaic, is calculated starting with the DEM with the smallest date difference to the validation. The third
approach hereafter named "DEM time series" is based on the extrapolation of elevation values using the entire or a subset of
the DEM time series. For this latter approach, groups developed several solutions to derive the elevation change over time.
These solutions differ in the use of single pixels or elevation bins as well as different temporal linear regressions to extract the
elevation trend.

### 3.3.2 Spatial bias correction

Spatial trends and vertical deformation resulting from sensor behaviour, data acquisition, and processing can systematically
bias relatively large fractions of the DEMs. These large-scale spatial trends at a scale at a horizontal scale of hundreds of
metres up to a few kilometres develop preferentially in the along and across-track directions. In TanDEM-X DEMs, these
trends are called "ramps" and appear as constant offsets of multiples of the height of ambiguity. Such linear trends are caused
by errors in the unwrapping of the interferometric phase ("phase jump"). Biases in ASTER DEMs caused by uncontrolled
satellite movements ("jitter"; e.g., Ayoub et al., 2008) manifest as structured noise in the form of undulation with an amplitude
of about 5 m and a pseudo period of several kilometres.
To correct the ramps in the TanDEM-X, some participants (DLR, FAU) fitted planes or higher-order polynomial functions on
stable terrain, which were then subtracted from the entire elevation change grid. Similarly, UGA removed a first-order spatial
polynomial vertical bias in ASTER DEMs. ASTER undulations were corrected by some groups (ETH, UIO) by fitting a sum
of sine functions during the DEM generation as proposed by (Girod et al., 2017). Additionally, one group (LEG) applied a
correction for the across and along-track shifts in ASTER DEMs inspired by Gardelle et al. (2013).

### 3.3.3 DEM co-registration

Remaining systematic errors in the DEMs, such as shifts and rotations resulting from georeferencing techniques and processing
distortion, need to be minimised through co-registration processes before performing DEM differencing. These errors are
visible on stable terrain, i.e., the stable bedrock off-glacier terrain, where elevation change is expected to be zero when no
geomorphic changes occur within the study period. Co-registration involves the selection of a) stable terrain, b) a reference
DEM for the estimation of the transformation (e.g. translation, rotation) between the raw DEMs and the reference and c) the
co-registration algorithm to align the raw DEMs.
In the experiment, groups adopted various co-registration procedures. Most groups used areas outside the RGI glacier outline
as stable terrain (e.g., BAW, DLR, FAU), but they also buffered the glacier outline to exclude periglacial areas affected by
mass movements. Some groups (e.g. UGA, UIO, USG) additionally excluded steep areas (greater than 20 to 50 degrees) from
their stable areas. The Copernicus DEM was the primary reference DEM for most groups, while others (DLR) included the



TanDEM-X global 90 m DEM (González et al., 2020). One group (LEG) applied a co-registration between two ASTER DEMs using the older one as a reference.

The DEM co-registration algorithm widely used in the experiment, and glacial studies in general, is the algorithm from Equation 3 of Nuth and Kääb (2011). Various implementations in different repositories have been used, including "demcoreg"

(Shean et al., 2023) and "xDEM" (Xdem Contributors, 2021). This algorithm iteratively estimates translation using the slope and terrain aspect in the regression model. It effectively removes horizontal and vertical shifts but does not correct for rotation or scaling between DEMs. Another group (UIO) used a least-squares matching algorithm implemented in the OPALS tool (Pfeifer et al., 2014) to estimate the full 3D affine transformation parameters of the DEMs from the reference one. Furthermore, two groups (UST, LEG) used the approach developed by Berthier et al. (2007). Any remaining biases after co-registration

were either ignored if they were smaller than the pixel resolution or corrected in the z-direction.

### 3.3.4 Noise filtering and void-filling

ASTER and TanDEM-X DEMs are both affected by artefacts, which are small-scale noise at a horizontal scale of typically tens to hundreds of metres. These artefacts are generated during the DEM processing due to sensor characteristics, surface properties, and topographic factors. For ASTER, artefacts such as sinks and bumps occur in regions with low image contrast

(e.g., saturation on the bright snow surface) resulting in poor image correlation. This, as well as the presence of clouds, also causes voids in the ASTER DEMs. Artefacts in TanDEM-X DEMs are produced by phase unwrapping errors. Additionally, in mountainous regions, the InSAR acquisition geometry (side looking) leads to shadowing and layover by steep slopes and vertical cliffs, which generate artefacts and voids. Once the DEMs are co-registered, these artefacts can be detected and removed.

Filtering approaches adopted by the participants were primarily based on statistical parameters like standard deviation, normalised median absolute difference (NMAD), or threshold values to remove gross errors. Noise filters and consequent void-filling were applied to the DEM, i.e., to the elevation change or to the elevation change rate maps. For example, DEM pixels that showed an absolute elevation difference from the reference DEM greater than a vertical elevation threshold were removed (e.g. BAW, UIO). This was applied per single pixel, within a certain radius (Hugonnet et al., 2021), by some groups

(e.g. ETH, LMI) or for elevation bands by other groups (e.g. GAC, UGA, UZH, LEG). Other filter solutions removed pixels with unrealistic absolute elevation change rate values (e.g., LEG, UZH) or elevation change (e.g., UST).

Voids in the DEM difference were filled by almost all groups using spatial methods such as the local or global hypsometric approach (McNabb et al., 2019). This method divides the glacier into elevation bins of typically 50 m or 100 m intervals and assigns the average elevation change of the corresponding elevation bin to the data voids. The UIO group also applied a simple

Inverse Distance Weighting interpolation.





### 3.3.5 Radar penetration correction

The InSAR signal can penetrate through the glacier surface, introducing a vertical bias. The specific penetration depth is related to the wavelength and varies locally and temporally depending on the conditions of the glacier surface during data acquisition (Dall et al., 2001; Rignot et al., 2001; Dehecq et al., 2016), making it difficult to correct. Typically, InSAR penetration depths are large for dry and cold snow and ice and lower for temperate snow and ice, and increased moisture content (Rignot et al., 2001). For TanDEM-X, the penetration depth is within a few metres, but average penetration depths of up to several metres (> 5 m) have been reported previously (Abdullahi et al., 2018; Dehecq et al., 2016; Li et al., 2021; Bannwart et al., 2023), while for the longer wavelength C-band of SRTM, even higher penetration depths can occur (Dall et al., 2001; Rignot et al., 2001).

In the inter-comparison experiment, the penetration bias was corrected only for the Baltoro study site by two groups, while others included it in the error budget (Sect. 3.3.7). The UST group captured penetration within their probabilistic framework as an elevation-dependent Gaussian probability distribution, as proposed by (Agarwal et al., 2017). Similarly, the GAC group applied an elevation-dependent C-band penetration model to the SRTM dataset based on the specific results for the East Karakoram region by Kumar et al. (2019). The X-band radar penetration model was applied to the TanDEM-X tiles collected in January and February based on the C/X-band penetration differences calculated for the Karakoram region by Lin et al. (2017). On a larger regional scale, such an assessment based on SRTM X/C-band data needs to consider also the differences in processing of the two data streams.

### 3.3.6 Temporal corrections

Homogenization of the observation period and thus temporal correction of the glacier elevation change estimate may be required when comparing different datasets, such as for IPCC estimates. In this experiment, such a correction was required when the spaceborne observation dates differed from the airborne validation period (study sites Hintereis, Aletsch and Vestisen).

The participants followed different strategies for temporal corrections: (a) no temporal correction (e.g., FAU, UIO, USG), which assumes that the observed elevation change rate corresponds to change rate of the validation period, (b) linear scaling using the long-term trend (e.g., BAW, UGA, UIO), based on the same assumption, (c) annual corrections using glaciological observations (e.g., LEG, UZH, UIO), and (d) non-linear regressions of the experiment DEM time series (ETH, LMI), which aimed at correcting for the long-term trend as well as for annual and seasonal variabilities.

The corrections applied by the participants still resulted in remaining temporal differences from the validation dates ranging from a few days to more than a year. To correct for these remaining temporal differences, we used a simple approach by Zemp and Welty (2023) that temporally downscales seasonal observations of glaciological mass balance using sine functions, and adjusted spaceborne elevation change results (see Sect. 2.6). Corrections of (remaining) temporal differences between the





spaceborne elevation changes and the validation dates ranged from zero to 2.5 m, depending on the differences from the start and end validation dates.

## 3.4 Uncertainty assessment

The overall error of a glacier-wide elevation change from DEM differencing originates from the input data, DEM processing steps, and corrections. In the experiment, a large set of different (and sometimes incomparable) methods were used by the participants, including different error contributions and metrics to calculate the dispersion of the distribution (e.g., NMAD and standard deviation). The overall uncertainty was calculated by the participants, assuming the independence of the different error sources (quadratically summed), and reported at a 95% confidence interval.

Participants considered the following error sources:

- Pixel elevation change error, propagated to the mean elevation change, considering or not considering spatial correlation. The UST group relied on a Bayesian probabilistic approach to the problem and calculated the uncertainty of elevation change as the 90% credible interval of the posterior probability distribution function (Guillet and Bolch, 2021). Other groups (e.g., LEG, LMI) followed the approaches described by Berthier et al. (2016), Magnússon et al.

(2016), and Wagnon et al. (2021). Some groups (e.g., BAW, FAU, UGA) also included spatially correlated errors based on the method by Rolstad et al. (2009), while others (ETH, UGA) considered both spatially correlated errors and heteroscedasticity (i.e. variability in the amplitude of errors, for instance explained by terrain slope or quality of stereo-correlation for DEMs) in the same framework (Hugonnet et al., 2021, 2022).

- Errors in glacier outline and area mapping. Area uncertainty was quantified by buffering the RGI outline (ETH) or

assuming a 10% uncertainty of the RGI inventory (LEG).

- Errors due to missing observations. The errors related to the interpolation of missing values were quantified as five times the uncertainty of the measured pixels (LEG, UGA), as proposed by Berthier et al. (2014) and Brun et al. (2017).

- Error related to temporal mismatch or temporal correction (e.g., BAW, GAC).

- Error related to radar penetration or radar penetration correction (if applied). The uncertainty due to correction was

estimated by the GAC group for the C-band and X-band as 1 m and 4 m, respectively, while BAW included the signal penetration in the error budget.

## 4 Results

## 4.1 Implementation of glacier elevation change assessment

In general, participants used the provided DEMs (70% to 90%, depending on the study site) rather than self-processed DEMs

and preferred working with pairs (or sometimes mosaics) of DEMs over time series (Fig. 6). This may indicate that time series





approaches are not widespread in the community. Time series approaches were not applied in any of the radar results, but mosaic solutions were employed especially for the larger regions.

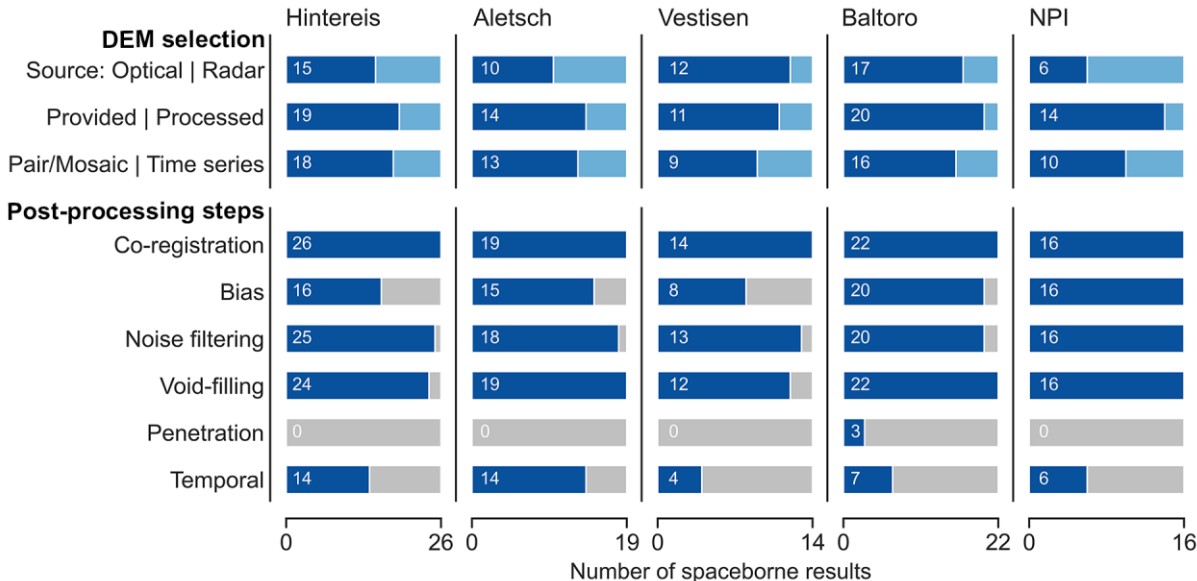

**Figure 6: Overview of submitted spaceborne results. The number of results according to DEM selection options (option 1 | option 2, i.e. dark blue bar | light blue bar) and post-processing steps (dark blue bars) for each study site. Grey bars indicate no results. Note that 'Mosaic' is included within the 'Pair/Mosaic' DEM selection option. For Baltoro and Northern Patagonia Icefield (NPI), the 'Radar' option includes the number of results that combine optical and radar. In addition, numbers for Baltoro and Northern Patagonia Icefield refer to reference results only, i.e. the full suite of results from this sensitivity experiment, where steps in the processing workflow were toggled on/off, are not included.**

All participants co-registered the DEMs, while spatial bias correction was applied differently depending on the site and sensor. Almost all participants proceeded with noise filtering and void-filling. Radar signal penetration, which is relevant when at least one DEM is derived from a radar sensor, was corrected by two groups for three out of five radar results of Baltoro, but for none of the radar results of the Northern Patagonian Icefield. Temporal corrections were applied to only about half of the submitted results by the participants, but have been complemented by temporal corrections to fit the target periods for the first experiment (Sect. 3.3.6).

The sensitivity study at Baltoro and at the Northern Patagonian Icefield, toggling each post-processing step on and off to quantify its impact on the reference result, received limited contributions. Only one out of four groups provided a sensitivity study for the Northern Patagonian Icefield. Therefore, the sensitivity study could only be conducted for Baltoro, where almost all submissions for both periods provided the mean elevation change without co-registration. In addition, half of the results came with separate contributions of bias correction, noise filtering, and void-filling. The impact of temporal and penetration corrections on the reference results could only be quantified by a few groups. Spaceborne results and related processing steps are available for Hintereis (Table S16), Aletsch (Table S17), Vestisen (Table S18), Baltoro (Tables S19 and S20 for periods 1 and 2, respectively), and Northern Patagonian Icefield (Tables S21 and S22 for periods 1 and 2, respectively).





## 4.2 Elevation change assessment for glaciers with airborne validation data

In the first experiment, participants calculated glacier elevation changes of Hintereis, Aletsch, and Vestisen and their estimates were compared to the airborne validation data. Figure 7 shows the resulting elevation-change maps of Hintereis from the 26 submitted results and the validation data. The results from the airborne surveys show a glacier-wide mean elevation change of $-10.6 \pm 0.3$ m (95% confidence interval) between 8 October 2010 and 21 September 2019 with the largest elevation changes of up to $-50$ m over the glacier tongue, a decrease in ice loss with elevation, and close to zero changes in the accumulation

area. The same elevation-change pattern is noticeable in the spaceborne results. Off-glacier, the impacts of DEM selection and post-processing strategies are evident. As such, the elevation differences feature different spatial coverages, as well as different patterns, and ranges of noise. Also, different levels of filtering both on and off-glacier become visible. Note that the airborne elevation-change map covers the full glacier extent over the validation period, but not the lowest part of the glacier tongue as mapped in 2003 by RGI 6.0 (see Table S1 and Fig. S1 for details). Nonetheless, any potential positive bias in the spaceborne

results, resulting from averaging glacier-wide elevation changes that include the glacier forefield within the RGI outline, is offset in this test site by proglacial sediment erosion during the validation period.

The elevation-change patterns of Aletsch (Fig. A5) follow the general trend observed in Hintereis. However, the level of voids and noise, as well as remaining bias both on and off-glacier is more pronounced in the spaceborne maps, likely due to the larger study site. These features are more visible in the ASTER results using the DEM pair and mosaic approaches than in the

time series approach. The TanDEM-X results show different signals in the large accumulation area, with several results indicating more negative elevation changes than those observed in the airborne dataset, although these were considered low confidence.

For Vestisen, the provided ASTER DEMs suffer from large data voids and thus low spatial coverage on the glacier with data voids in the accumulation area over the entire time series (white area in Fig. A2). This resulted in corresponding data voids in

the elevation-change maps or extensive interpolation (Fig. A6). Consequently, the workflow applied in other sites failed to yield satisfactory results − more than half of the optical outcomes were flagged as low confidence results. The elevation change maps from the self-processed ASTER DEMs exhibit better agreement with the airborne result, but considerable noise is still evident. Conversely, the two TanDEM-X results display remarkably smooth elevation-change maps, but the signal indicates less negative changes than the airborne result. However, we note the temporal difference of three years between the TanDEM-

X results and the validation data.





**Figure 7: Compilation of elevation change rate maps in metres per year for Hintereis.** The spaceborne results, sorted by the type of sensor (optical and radar) and by DEM selection strategies (e.g., pair, mosaic, and time series approaches) are shown together with the airborne validation result. Group labels are represented by three-letter acronyms with corresponding result numbers (#). Results flagged as low confidence are indicated by group labels in *italic*. Glacier outlines are from RGI v6.0 (RGI Consortium, 2017), showing the target and neighbouring glaciers with thick and thin outlines, respectively. Note that UIO-3 and UIO-4 do not show stable terrain as they used a time series approach based on elevation bins (cf. Table S12).

The mean glacier-wide elevation changes (dh) and elevation change rates (dh/dt) are shown for Hintereis in Fig. 8. The spaceborne results are almost equally distributed around the validation data but come with a spread in mean elevation change of about 10 m or in mean elevation change rate of 1 m per year. The large spread remains, even when excluding low confidence





results (dotted lines in Fig. 8). A closer look at the two left subplots – showing calculated elevation changes and corresponding change rates – confirms that the differences to the validation data (and corresponding rankings) of the spaceborne results cannot be compared directly due to differences in the survey period. For instance, the runs by FAU-1 (with a longer survey period) and FAU-2 (with a shorter survey period) switch their ranking when elevation changes and elevation change rates are

compared. Similarly, the run of UGA-1 features a more negative elevation change but a less negative elevation change rate due to the long survey period (compared to validation). For those spaceborne results that did not match the validation period (i.e., horizontal lines longer or shorter than the target period), a temporal correction was applied (Sect. 3.3.6) and the corrected elevation change and change rate are shown in Fig. 8c. We note that temporal corrections homogenise the results to a common time period, but do not reduce the overall spread between the results.

The results for Aletsch (Fig. A7) and Vestisen (Fig. A8) show a similarly large spread of the spaceborne results and a similar need for temporal correction to facilitate direct comparison with the validation data. In the case of Aletsch, we note that the majority of the spaceborne results are more negative than the validation, which might indicate a possible bias in the airborne data. The challenges of Vestisen resulted in a large spread of the spaceborne results, with some strong outliers flagged as low confidence.

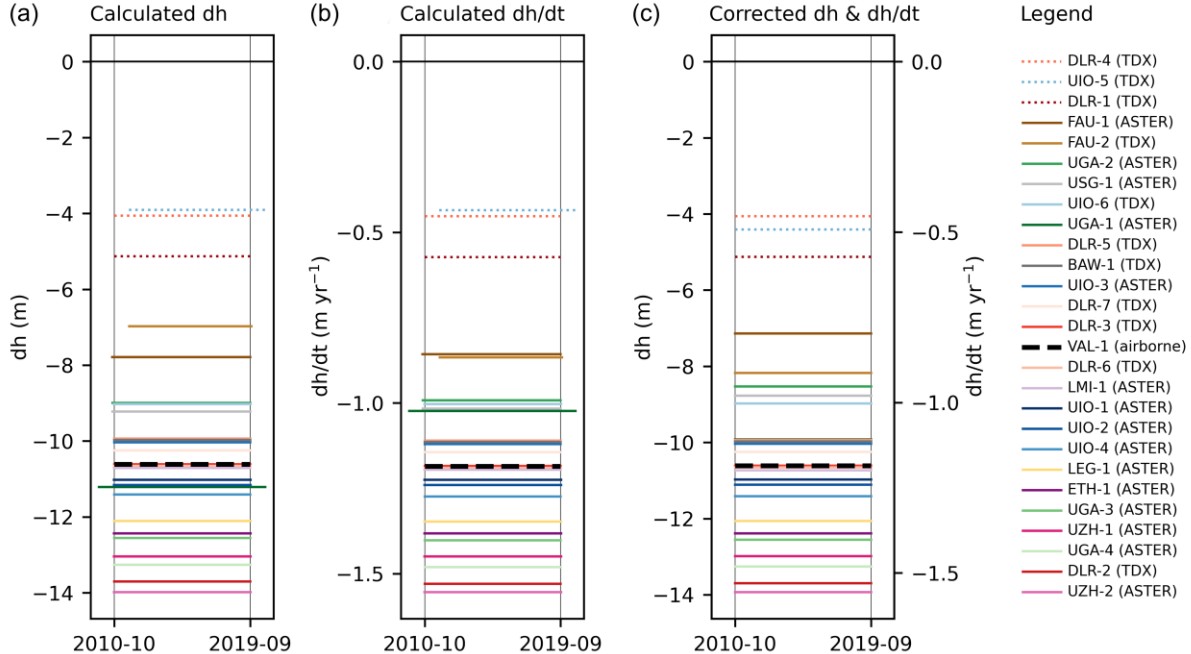


**Figure 8: Elevation changes of Hintereis from airborne validation and spaceborne results. The results are shown as originally calculated (a) mean elevation changes (dh) and (b) corresponding elevation change rates (dh/dt), as well as (c) after temporal corrections to the validation period, for contributions where this was not done by the participants. The spaceborne results are labelled for group and result number, with DEM sources in brackets. Low confidence results are marked by dotted lines. The**

**airborne validation (VAL-1) result is shown as a dashed black line with the corresponding validation period marked by vertical grey lines. Error bars of the validation data are smaller than the line width. Error bars of the spaceborne results are shown in Fig. 11. The legend is sorted by descending elevation change (rate) values (c), which corresponds to negative and positive differences to the validation result.**



### 4.3 Regional-scale elevation-change assessment and sensitivity study

The second experiment focused on the larger regions of Baltoro and the Northern Patagonian Icefield for two target periods, 2000−2012 and 2012−2019. Instead of comparing spaceborne elevation-change results with airborne validation data, this experiment focused on parsing the effect of various steps within the processing chains used by each group.

Similar to the other study sites, the elevation change rate map for Baltoro displays a wide range of results (Fig. 9). By visually examining the maps, a surge-type pattern can be observed for a tributary from the north. However, some submissions show

completely opposite patterns of spatial elevation change, either at the glacier tongues or the central part of the Baltoro. Such opposite signals are not observed for the second observation period, but significant differences in the magnitude of change are still noticeable. Some results exhibit remarkable negative values, particularly in the southern or central part of the study region. In these cases, there also appear to be stronger offsets off-glacier, which is also partially observed in the first period.

The large spread on the calculated elevation differences is also reflected in the mean elevation change rate for both periods

(Figs 10a and 10d). In the first period (2000−2012), the results range from positive to slightly negative elevation change rates, while in the second period (2012−2019), almost all the results show negative values. It is important to note that the values and magnitudes cannot be directly attributed to a specific input data source since participants used different combinations of DEMs − i.e., ASTER/ASTER, TanDEM-X/TanDEM-X and for the first period also SRTM/ASTER and SRTM/TanDEM-X − resulting in different observation periods. Furthermore, two results did not cover the entire study area and participants

employed a wide variety of approaches, although many used mosaicking techniques due to the large size of the region.

In the case of the Northern Patagonian Icefield, the limited number of submitted results prevents a quantitative comparison. Out of a total of eight estimates for each period, only four covered the entire icefield, and only two provided measurements for the central part of the icefield (Figs. A9, A10). However, it is still possible to compare a subset of 55 glaciers observed by the participants in the eastern part of the Northern Patagonian Icefield, where favourable cloud conditions allow more

observations from optical images. The elevation change rate maps show a more consistent pattern as compared to Baltoro and there is no clear trend indicating higher or lower estimates based on the type of data used (i.e., radar or optical). Moreover, in all results, the change rates appear to be more negative in the second observation period.



**Figure 9: Compilation of elevation change rates maps in metres per year for Baltoro for the two target periods and from the different DEM source combinations, i.e. ASTER/ASTER, SRTM/ASTER, SRTM/TanDEM-X, and TanDEM-X/TanDEM-X. Group labels are represented by three-letter acronyms with corresponding result numbers (#). Results flagged as low confidence are indicated by group labels in *italic*. Glacier outlines are from RGI v6.0 (RGI Consortium, 2017), showing the target and neighbouring glaciers with thick and thin outlines, respectively.**





As part of the inter-comparison experiment, the Baltoro study site was also used to assess the impact of each post-processing
step and correction on the mean elevation change for both target periods (Fig. 10). Figures 10a and 10d show the values of the
reference result, while Figs 10b and 10e reveal the impact of forgoing each individual step. While the small sample size of the
sensitivity studies might limit the significance of the results, it still illustrates the relevance of the different post-processing
steps. The strongest influence on the results comes from the co-registration in most processing chains. The magnitude of the
correction is not consistent per group or approach, and the direction is different for the two periods. The influence of other
processing steps is an order of magnitude lower on the overall dispersion of the results. In general, their impact on the reference
result is smaller than the one of co-registration, but can still be crucial in individual cases. For example, void-filling can be
essential to remove a bias in the results for DEMs with larger data gaps. In the second period, void-filling had a smaller impact,
which indicates larger data gaps in the first period, partly related to the use of SRTM. Radar penetration corrections were only
conducted by a few groups, either because no radar product was used or because it was not feasible to estimate the related bias
within this experiment. Instead, this component was included in the error budget. A temporal correction was performed by
only one group (GAC-1) and shows the smallest contribution to the overall elevation change results, likely because this group
matched well to the target period (Figs. 10c and 10f).



**Figure 10:** Elevation change rates (a, d) and related corrections (b, e) for Baltoro for the periods 2000−2012 (upper panels) and 2012−2019 (lower panels). The REFERENCE results from the full processing chain (i.e. including all applied corrections) are shown with the individual corrections (Δ) related to co-registration (COR), bias (BIAS), filtering (FILT), void filling (VOID), radar penetration (PEN), and temporal correction (TEMP). Individual spaceborne results and target periods are shown in panels (c) and (f). DEM sources are indicated by point markers as shown in the legend of the panel (b). The box plot represents the mean and the interquartile range, including all results. Note the different scales of the y-axis for the two target periods and some error bars outside the plot range.



## 5 Discussion

### 5.1 How reliable are spaceborne estimates of glacier elevation changes?

Spaceborne optical and radar missions have opened the opportunity to observe elevation changes of all glaciers worldwide.
Among the numerous sensors available (Berthier et al., 2023), ASTER and TanDEM-X have been among the most important for estimating glacier changes from DEM differencing at regional to global scales (e.g., Brun et al., 2017; Braun et al., 2019; Hugonnet et al., 2021). The present study is the first systematic inter-comparison experiment to assess results from different research groups and workflows based on common, predefined glaciers, input data, and target periods. Thereby, the validation of these results using airborne surveys – which feature optimised acquisition dates for glacier applications, high spatial
resolution, and overall quality – provided valuable insights into the accuracy of glacier elevation change from spaceborne data. Across all experiments and sites, we found a large spread in elevation change results of more than 1 m per year, which poses a substantial challenge for reconciling glacier change rates from different studies (Zemp et al., 2019; Hugonnet et al., 2021). Also, we found large differences in reported uncertainties often not overlapping with airborne validation data. At the same time, results showed that ensemble estimates (Table 1, 2) are promising to reduce random errors from different instruments
and processing methods, but still require a more comprehensive investigation and correction of systematic errors. Figure 11 illustrates these findings with the example of Hintereis. The deviations of individual spaceborne results from the validation data range from close to zero to positive and negative differences of up to 0.5 m per year or more, with reported uncertainties (at a 95% confidence interval) ranging from about ±0.1 m per year and more than ±1.0 m per year. It is worth emphasising that uncertainty estimates vary considerably due to the different methods (e.g. stable terrain selection, statistic measure, spatial
auto-correlation) and types of error sources (e.g. area, area changes, gap filling, etc.) considered by the groups, as discussed in Sect.3.4 and elaborated in their supplementary table.

The ensemble mean of the full experiment sample perfectly fits the validation result while the ensemble means of the ASTER and TanDEM-X underestimate and overestimate, respectively, the validation data, although within the standard error of the corresponding sample.



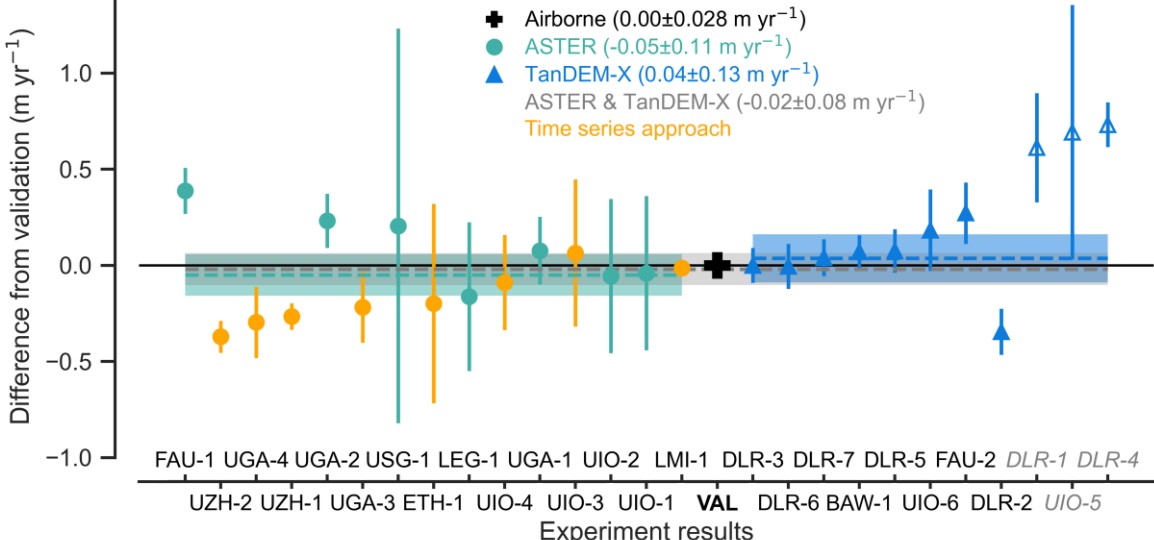

**Figure 11: Difference of the spaceborne results from the airborne validation data for Hintereis. Spaceborne results are shown including temporal corrections (cf. Fig. 7c). Low confidence results are marked by empty symbols and by corresponding labels in *italic* and grey. Results based on a time series approach are highlighted in orange. Reported uncertainties are shown as vertical lines. Note that the uncertainty of the validation result is smaller than the marker. Ensemble means (dashed lines) and related 1.96 standard errors (shadings) are shown and corresponding values are given in the legend, excluding low confidence results.**

The experiment results from Aletsch (Fig. A11) and Vestisen (Fig. A12) show the same general picture with the exception that in these cases the ensemble means deviate from the validation data with differences between 0.3 m per year and −0.2 m per year. While we cannot exclude that these biases at least partly originate from the airborne validation data, it is clear that the large spread between the experiment results represents a major challenge for reconciling estimates of glacier elevation change that rely on spaceborne data and different workflows.

We did not find if any approach performs best in all cases, due to our small sample size of three validation sites and due to the large range of reported random errors. As such, results from both ASTER and TanDEM-X rank amongst the closest as well as amongst the most distant from the validation data (Figs. 11, A11, A12). Thereby, participants followed different strategies in selecting DEMs. For pair and mosaic approaches, optical data were usually selected from the end of the summer months, whereas for radar data winter DEMs were preferred. DEM differencing from time series analysis – as applied by ETH, UGA, UIO, and UZH – is a promising approach for optical data to deal with spatio-temporally scarce elevation data but did not clearly outperform (i.e., show stronger agreement with validation data) other approaches (Figs. 11, A11, A12). We note that none of the groups used a time series approach with TanDEM-X data, partly to avoid issues with radar penetration, which is expected to have a larger impact in cold winter than in wet summer snow conditions. One possible explanation for the variable performance across processing steps is that all approaches perform well in cases with good-quality DEMs available close to the target period that covers the glacier extent. In the absence of good-quality, spatially extensive DEMs close to the validation period, workflows and results are more heavily influenced by researcher decisions regarding DEM processing, selection, and post-processing. Finally, we note that our sample with validation data from three study sites can only provide initial, qualitative



insights. A much larger, worldwide sample of high-quality airborne or spaceborne benchmark datasets would be required for
a quantitative evaluation of best practices.

Among the various approaches employed by the groups participating in this inter-comparison experiment, Hugonnet et al.
(2021; labelled "ETH" in this study) have developed a fully automated workflow for processing and post-processing the entire
ASTER archive, and have also published elevation change time series for almost all glaciers globally. The experiment results
show strong agreement with airborne validation data, confirming that the approach by Hugonnet et al. (2021) can currently be
considered good practice for computing glacier elevation changes from spaceborne DEM differencing. As such, their DEM
processing routines (Girod et al., 2017) are well tailored to ASTER stereo images, their time series approach is able to derive
a maximum of information from DEM stacks, and their post-processing workflow seems to optimally derive glacier-wide
results and related uncertainties from off-glacier terrain. Their experiment runs, which were based on the published dataset,
generally perform well in the present study, rank one of the closest to the validation data in the case of Vestisen (Fig. A12),
and come with reasonable error bars, which overlap with validation data and ensemble means in all experiments.

**Table 1.** Ensemble mean elevation change rates (dh/dt) and 1.96 standard errors (σ) for Hintereis, Aletsch and Vestisen together with the validation results and related uncertainty. The ensemble values include the temporal corrections to validation periods (Figs 7c, A7c, A8c). Ensemble statistics are calculated excluding low confidence results and shown with corresponding sample sizes (#). Full data records for each group are provided in the supplement.


| Ensemble | DEMs sources | Hintereis | Aletsch | Vestisen |
|---|---|---|---|---|
| | | 08.10.2010 to 21.09.2019 | 13.09.2011 to 21.09.2017 | 02.09.2008 to 10.09.2020 |
| | | dh/dt ± σ (#) (m y$^{-1}$) | dh/dt ± σ (#) (m y$^{-1}$) | dh/dt ± σ (#) (m y$^{-1}$) |
| Subset | ASTER/ASTER | −1.24 ± 0.11 (15) | −1.37 ± 0.10 (10) | −0.40 ± 0.21 (5) |
| Subset | TanDEM-X/TanDEM-X | −1.15 ± 0.13 (8) | −1.30 ± 0.13 (3) | −0.08 ± 0.03 (2) |
| Full | all | −1.21 ± 0.08 (23) | −1.36 ± 0.08 (13) | −0.31 ± 0.19 (7) |
| Validation | Airborne | −1.19 ± 0.03 (1) | −1.14 ± 0.15 (1) | −0.36 ± 0.02 (1) |

**Table 2.** Ensemble mean elevation change rates (dh/dt) and 1.96 standard errors (σ) for Baltoro and Northern Patagonian Icefield (NPI) for two periods. Ensemble statistics are calculated excluding low confidence results and shown with corresponding sample sizes (#). For NPI, area-weighted results are reported for a subset of 55 glaciers in the eastern part, which were covered by all groups.
Full data records for each group are provided in the supplement.

| Ensemble | DEMs sources | Baltoro | | NPI (55 glaciers subset) | |
|---|---|---|---|---|---|
| | | Period 1 | Period 2 | Period 1 | Period 2 |
| | | 2000−2012 | 2012−2019 | 2000−2014 | 2014−2019 |
| | | dh/dt ± σ (#) [m y$^{-1}$] | dh/dt ± σ (#) [m y$^{-1}$] | dh/dt ± σ (#) [m y$^{-1}$] | dh/dt ± σ (#) [m y$^{-1}$] |
| Subset | ASTER/ASTER | 0.07 ± 0.22 (6) | −0.17 ± 0.14 (7) | −0.93 ± 0.15 (3) | −1.18 ± 0.31 (3) |
| Subset | SRTM/ASTER | −0.14 ± n.a. (1) | n.a. | −1.04 ± 0.08 (4) | n.a. |
| Subset | SRTM/TanDEM-X | −0.02 ± 0.15 (2) | n.a. | −0.78 ± nan (1) | n.a. |
| Subset | ASTER/TanDEM-X | n.a. | n.a. | n.a. | −1.44 ± 0.05 (4) |
| Subset | TanDEM-X/TanDEM-X | n.a. | −0.26 ± 0.39 (2) | n.a. | −1.66 ± n.a. (1) |
| Full | all | 0.02 ± 0.15 (9) | −0.19 ± 0.13 (9) | −0.97 ± 0.09 (8) | −1.37 ± 0.16 (8) |

## 5.2 Which processing steps impact elevation change estimates the most?

While the first experiment was carried out on relatively small glaciers and ice caps with the goal of estimating elevation
changes for a predefined target period, the second experiment aimed to use the same processing workflows at the much larger





Baltoro Glacier and over the entire Northern Patagonian Icefield. While one might expect random differences to average out

over larger regions, this experiment showed that a wide spread between the results of 0.5 m per year and more still exists (Fig. 10). This indicates that there are remaining systematic differences between the approaches, which have a strong impact on the results. Similar to the first experiment, we find a large range of uncertainty estimates ranging from a few decimetres to a few metres, often not overlapping with the ensemble means (Fig.10).

From both experiments, we found that there are three processing steps in the workflow that have a major impact on the results

across experiments:

(i) **DEM production**: Ideally the DEMs are produced within a workflow with optimised routines for glacier applications. As such, the settings of saturation thresholds, cloud detection, and image selection can have a strong influence on the quality of ASTER DEMs over glaciers. Similarly, a priori knowledge of the topography is important for correct phase unwrapping over changing glacier surfaces in the case of TanDEM-X. In practice as well as in our experiment, however, not all groups have

corresponding tools and expertise to generate spaceborne DEMs and, hence, relied on provided DEMs.

(ii) **DEM selection**: The selection of the best elevation data for a glacier change assessment depends on various factors related to spatial and temporal coverage, as well as to the DEM quality. Depending on available data, different strategies might be appropriate. As such, a pair approach with two high-quality DEMs, with few data voids, acquired at the end of the glacier ablation period provides maximum control during the processing workflow. However, if the 'perfect' DEMs are not available

or if a corresponding manual search is not feasible (e.g., for global applications), the mosaic or time series approaches can help to derive additional information from DEM stacks but have to deal with the temporal spread of corresponding elevation information.

(iii) **DEM co-registration**: Spatial alignment of DEMs is the fundamental precondition for any elevation change assessment. Different tools, in combination with priori or posteriori bias corrections, were used to correct shifts, rotations, and scale effects

between DEMs. Figure 10 illustrates that corresponding corrections can be up to a few metres per year and, hence, orders of magnitude larger than the long-term glacier change rates. We note that while the magnitude of the correction might be a first indicator of related uncertainty, the final accuracy of the correction depends on the skills of the approach to detect and correct spatial misalignments between DEMs.

The impact of these three processing steps is illustrated for Vestisen (Fig. 12), using results from UIO-1 and UIO-2 based on

ASTER DEMs. This example demonstrates that using a tool specifically developed for the ASTER sensor (e.g. MMASTER) to generate the DEMs significantly increases the spatial coverage from 33 to 98 % of the glacier surface, leading to a difference in the mean elevation change of several metres between the provided and self-processed DEMs (e.g. Fig. 12a vs. Fig. 12e). Remarkably, even when using the same self-processed or provided DEM pair, the use of two different co-registration methods – i.e., xyz-shift corrections (Nuth and Kääb, 2011) and least squares matching (Pfeifer et al., 2014) – that resolve different

transformation parameters can result in noticeable differences in the mean elevation change, as is evident from the comparison between e.g. Fig. 12e and Fig. 12f.



Compared to these three basic processing components of the workflow, other steps seem to have a smaller impact in general (Fig. 10). Still, they can be important for individual cases and depending on the use of optical or radar data. Here, we discuss a few issues in more detail with selected examples from the experiment runs.

**Spatial data voids**: DEMs from ASTER can suffer from large data voids in cloud- and snow-covered areas, such as in the accumulation zones of Vestisen or of the Northern Patagonian Icefield. While TanDEM-X is less impacted by clouds, it still can come with data voids due to radar layover and shadow effects in mountainous terrain. Participants have applied various void-filling techniques to calculate glacier-wide elevation changes. Unsurprisingly, the choice of technique impacts the final results, which depends on the ratio and spatial distribution of the data voids (McNabb et al., 2019; Huber et al., 2020). In the

example of Vestisen, the presence of multiple voids leads to a difference of about one metre in mean elevation change when using the local versus global hypsometric approach (Fig. 12c vs Fig. 12d). In contrast, when the DEM differencing shows fewer voids (Fig. 12g vs Fig. 12f), the two void-filling approaches have little impact on the mean elevation change values.

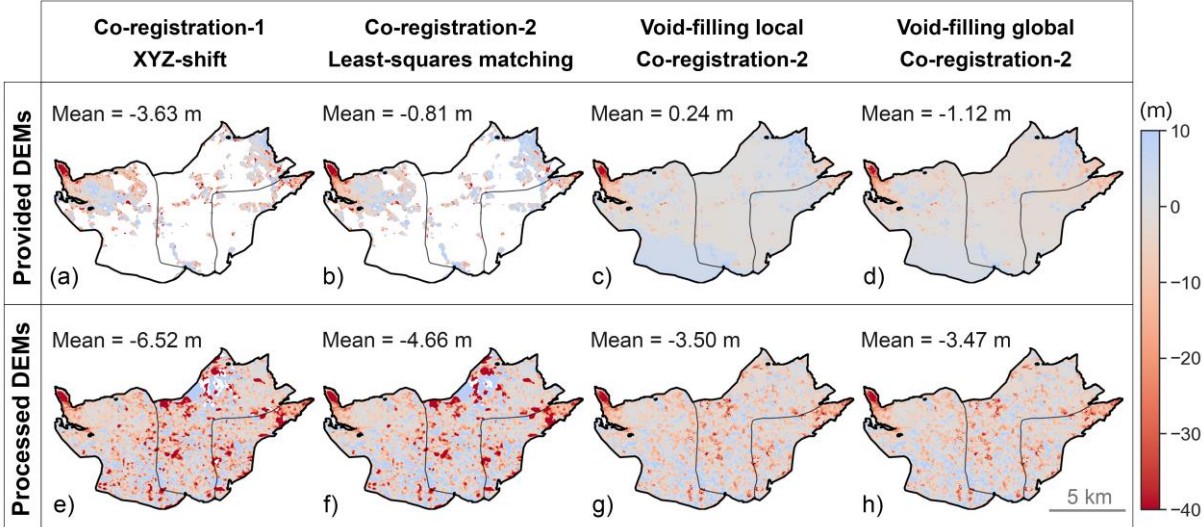

**Figure 12: Comparison of the elevation change maps of ASTER DEMs (DEM-pair approach) for Vestisen using provided DEMs**
**and self-processed DEMs, two co-registration algorithms (i.e., xyz-shift corrections by Nuth and Kääb (2011) and least-squares matching by Pfeiffer et al. (2014)) and two hypsometric approaches for void-filling (i.e. local and global). The maps c), d), and h) correspond to the results of UIO-1, UIO-2, and UIO-7 (Table S12). Note that void-filling is performed after filtering.**

**Differences in survey periods**: Participants applied three strategies to account for temporal differences between survey and target periods: no correction due to DEM selection close to target dates, corrections filling temporal gaps with annual
glaciological observations, and corrections based on long-term annual trends derived from the selected DEMs. Thereby, long-term trends were scaled by full years and decimal years to correct for annual and monthly differences, respectively. For the first experiment, we added temporal corrections to the results with remaining temporal differences to the validation dates (Sect. 3.3.6). The opportunities and limitations of these approaches are illustrated in Fig. 13. For survey dates with only a few days difference to the validation dates (at the end of the ablation season), temporal corrections are minimal and might be ignored in
view of observational uncertainties. For survey dates with differences of one or several hydrological years, a correction with




glaciological data (i.e., blue line at the end of the hydrological year in Fig. 13) can provide the annual corrections (after density conversion, cf. Huss, 2013). An annual correction based on the long-term trend from the DEMs (i.e., the black line at the end of hydrological years in Fig. 13) only works if the annual change corresponds to the long-term trend. In the example of Fig. 13, however, this is only the case in October 2011 (i.e., when the blue and black lines match). For the other three years (i.e.,

Oct 2009, Sep 2018, Sep 2020), an annual correction still results in an error of about 0.5 m. For survey dates that deviate from the annual minima, the required corrections can be substantial reaching up to a few metres, depending on the glacier's mass-balance amplitude (Braithwaite and Hughes, 2020). In such cases, long-term trends scaled by decimal years are not able to provide appropriate corrections but analytical (Zemp and Welty, 2023) or numerical (e.g., Huss et al., 2015) modelling might be required. In general, we can conclude that the selection of DEMs close to the annual mass-balance minima (around the 1st

of October for northern mid-latitudes) is a good strategy to reduce the impact of differences in survey periods. DEM selection close to the annual mass-balance maxima (between March and May) can be an option but is expected to be more variable in dependence on the accumulation history, and more subject to material density uncertainty associated with the elevation change to glacier mass change conversion. In any case, the same season should be used for start and end dates in order to avoid a bias in the derived trend.

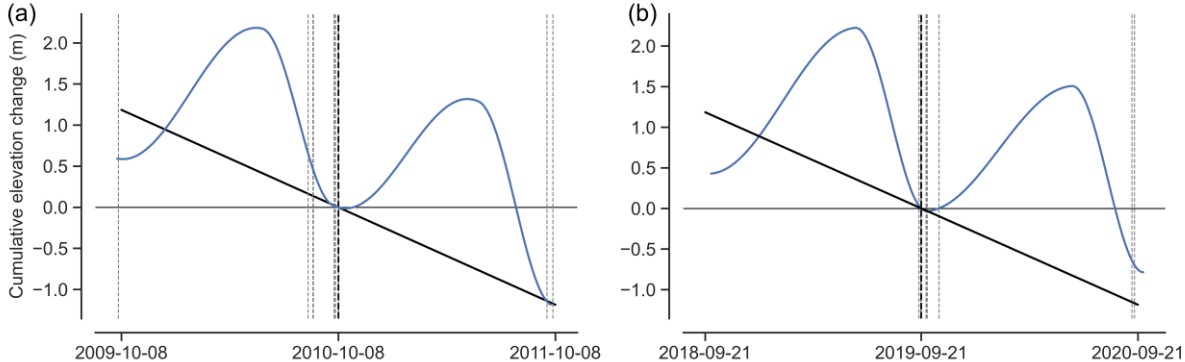


**Figure 13: Cumulative elevation changes around the start (a) and end (b) dates of the validation period for Hintereis. The seasonal course of cumulative elevation changes (blue line; averaging daily accumulation and ablation events based on Zemp and Welty, (2023)) and long-term trends from the validation data (black solid lines) are shown relative to the survey date between one year before and after the validation dates (vertical black dashed lines). The intersection of the survey dates from the experiment results**
**(vertical grey dashed lines) indicates the temporal corrections required to fit the elevation change of the validation period from 2010-10-08 to 2019-09-21.**

**Radar penetration**: For SRTM and TanDEM-X, a bias is potentially introduced through the penetration of the C- or respective X-band signal into snow and firn due to radar volume scattering (Dall et al., 2001). In our experiments, only two groups addressed this issue by applying radar penetration corrections, only for Baltoro; others included this in their uncertainty
estimate. The corresponding corrections were a few decimetres per year. A common assumption was that the radar penetration bias could be mitigated by selecting DEMs from the same season and, hence, with similar penetrations cancelling out as long as the radar frequency does not change. Using the provided TanDEM-X DEMs for Aletsch, we show the elevation differences between the summer (August) and winter (March) DEMs of the TanDEM-X in 2013 (Fig. 14), and thus the effect that an





inappropriate DEM selection could cause in alpine conditions. The negative difference below the median glacier elevation
(approximately 2300 m a.s.l.) is likely due to snow and ice melting on the glacier tongue. In contrast, the subsequent positive
bias (starting around 2800 m a.s.l.) may be explained by radar penetration into the winter snow and firn package. This bias,
which can be as high as 8 meters, aligns with the findings of a recent study by Bannwart et al. (2023) conducted in the same
location, while time series analysis for other regions even revealed higher seasonal signals (Vijay and Braun, 2017). These
results indicate that more research is urgently needed (Dehecq et al., 2016; Abdullahi et al., 2018; Li et al., 2021) on absolute
and relative radar penetration for different wavelengths − specifically with new bi-static spaceborne missions in view like ESA
Harmony (Lopez-Dekker et al., 2021).

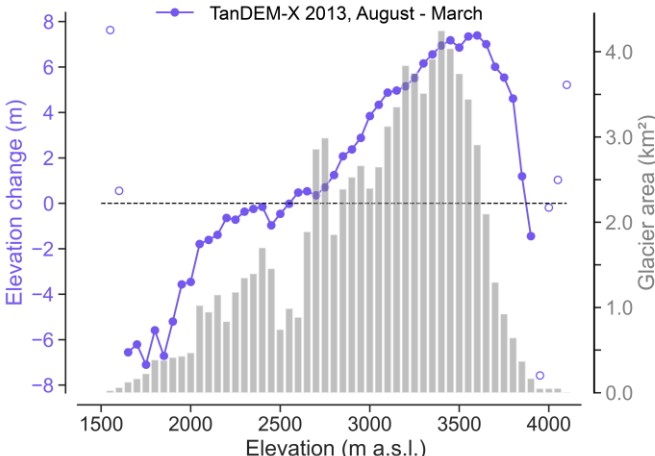

**Figure 14: The potential impact of radar penetration on Aletsch when two DEMs from different seasons are subtracted. Elevation
change of summer (2013.08.11) and winter (2013.03.21) DEMs from TanDEM-X are shown with the glacier hypsometry (grey bars,**
**second y-axis). For elevation bands that have a limited area (less than 0.1 km², the elevation change is shown as an empty circle.**

Besides the issues discussed above, any other step of the workflow as well as the auxiliary data can – under certain
circumstances – have a significant impact on the elevation change results. In addition, there are some issues that were not
covered in the present inter-comparison experiment but that can have an important impact on geodetic glacier mass change
assessments. As such two examples are discussed below.

**Glacier area changes** between the two geodetic surveys should be considered. In practice, regional assessments from
spaceborne sensors often use fixed glacier outlines from the latest version of RGI (RGI Consortium, 2017), with timestamps
ranging mostly between 2000 and 2010, assuming that the maximum glacier extent during the geodetic survey period was
within these outlines. If this assumption holds true, the calculated volume change within these larger outlines is correct (for
land-terminating glaciers without mass movements in the forefield), but the mean elevation change is expected to have a bias
due to limited or no ice loss in the glacier forefield (within RGI outlines). Also, there are cases where glaciers are larger than
mapped in RGI, such as due to surge (Sevestre and Benn, 2015) or tidewater dynamics (McNabb and Hock, 2014). As a
consequence, geodetic assessments should consider (i.e. map) glacier extents from both surveys (e.g., Berthier and Brun, 2019;
Sommer et al., 2020) when area changes are larger than a few percent.



**Density conversion** needs to be considered in a geodetic assessment when results are compared to glaciological observations
or when contributions to runoff or sea-level rise are reported. In practice, conversion factors are often just assuming bulk densities of about 900 kg m$^{-3}$, following Sorge's law (Bader, 1954), or 850±60 kg m$^{-3}$, based on Huss (2013). While these assumptions might hold for glaciers close to steady-state conditions and over longer time periods, conversion factors can vary substantially in cases of survey periods shorter than a decade, particularly if the observed average height change across the glacier is driven strongly by changes in accumulation zone snow and firn (Huss, 2013).

**5.3 What are the lessons learned from the inter-comparison experiments?**

In the present experiment, the participants aimed to estimate elevation changes from provided or self-processed DEMs from optical satellite stereo and spaceborne radar interferometry for predefined glaciers and target periods. As a result, we were able to compile and compare geodetic results from a large diversity of workflows currently employed by the scientific community. We found a large spread in glacier surface elevation change rates of more than 0.5 m per year, or more when considering low 785 confidence results, originating from differences across the entire workflow. Yet, it is worthwhile to note that a part of this spread might result from the experimental character of the study: groups tried to closely match the predefined target periods and used the opportunity to test various approaches that might not have been published outside the context of this experiment. In other words, based on the same input data, they would have published their elevation change estimates for a different period than the target period constrained partly by the availability of the validation data.

Ideally, an assessment of glacier elevation changes has full control over the geodetic workflow including DEM production, selection, post-processing, and corrections. And it comes with validation using high-quality DEMs from airborne or higher-resolution satellite data, such as Pléiades (Brun et al., 2017; Berthier et al., 2014), or Worldview (Noh and Howat, 2015).

More specifically, we can summarise the lessons learned following the general workflow (Fig. 4):

- Source: Each sensor has its strengths and limitations, which should be considered in view of the application for glacier change assessments. The combination of DEMs from the same source (optical or radar) allows for optimising the workflow as well as attributing and ideally solving differences between sources.

- Auxiliary data: The success of a workflow requires additional data, such as a reference DEM for the DEM production and co-registration, and glacier inventories for mapping glacier extent and area changes.

- DEM production: The photogrammetric/interferometric tools to produce DEMs from satellite images have a major influence on the performance of all subsequent steps of the workflow. Therefore, ideally, it should be carried out with software tailored to the sensor and optimised for glacier applications. Available DEMs need to be carefully and critically analysed with respect to their suitability for the purpose.

- DEM selection: Optimal selection of DEMs based on their quality and timing is essential for accurate detection of
long-term trends against noise and seasonal variability. Ideally, DEMs at the annual mass-balance minimum (i.e., at the end of the ablation season) are preferred to avoid issues with temporal corrections. In such a case, an approach



using DEM pairs of good quality can work perfectly well. Otherwise, mosaic or time series approaches can help to reduce the effects of random error associated with any individual DEM and extract additional information from the DEM stacks.

- Post-processing: All related steps aim at removing systematic biases from the result. Upfront, a good co-registration − ideally correcting for shifts, rotation, and scaling using well-distributed and stable terrain − is key. Thereby, strong corrections are an indication of potential remaining quality issues in the DEM. The relevance of other steps depends on case by case but might be essential, especially in the case of poor-quality DEMs.

- Corrections: Additional corrections, such as accounting for differences in survey periods and radar penetration, can
be relevant for the comparability of results and for correct trend detection.

- Results: Reports from different groups (or studies) need to provide full information, including survey dates, elevation change, interpolated area, glacier area, area changes, and meta-data on the workflow, in order to ensure full comparability and traceability.

- Uncertainties: Results need to come with sound uncertainty assessments, reported at comparable confidence levels.
Systematic biases are ideally quantified and corrected during or after post-processing, with corresponding uncertainties added to the error budget. Here, we see an urgent need for a coordinated community effort to establish consensus practices for assessing geodetic uncertainties covering error sources from all processing steps in a comparable manner.

- Reproducibility: Workflows should be entirely transparent and provided in the form of open-source code and scripts
that can be easily executed to reproduce results from any experiment. Community software development efforts such as xDEM are a critical step towards building consensus, formalising processing steps, and establishing best practices (Xdem Contributors, 2021).

In summary, we conclude that currently there is no single best, consensus method but several diverse, valid practices. Looking at the results from a corresponding ensemble can provide a robust mean as well as a realistic estimate of related uncertainties.
Yet the present experiment was limited to a few sites with validation data. A larger sample of benchmark datasets from airborne or high-resolution spaceborne sensors is needed for a comprehensive evaluation of different workflows and processing steps, e.g. for different co-registration approaches (Li et al., 2022; Pfeifer et al., 2014). A comparison of results between ASTER and airborne lidar and high-resolution satellites on about 500 glaciers (Hugonnet et al., 2021) indicated that a large amount of the differences to the reference data are random and, hence, might cancel out for estimates at larger scales. Furthermore, the
impact of seasonality and temporal corrections, area changes, and density conversion on their results warrants further investigation towards future assessments of glacier mass changes at regional to global scales.



## 6 Conclusions

In Spaceborne optical stereo and radar interferometry missions have opened the opportunity to observe elevation changes at high spatial resolution for all glaciers worldwide. The present study is the first inter-comparison experiment to evaluate the
results from different research groups and workflows based on predefined input data for given target glaciers and periods.

Across all experiments, we found a large spread in elevation change estimates of up to 0.5 metre per year or more. This corresponds to a spread of 5 m over a typical survey period of one decade. While this large spread might partly be attributed to the experiment setup, it still poses a challenge for individual results from spaceborne surveys to reliably detect current glacier change rates, which are in the same order of magnitude. On the other hand, the results also show that an ensemble
approach is promising to reduce random errors from different instruments and processing methods, but still requires a better investigation and correction of systematic errors. Reported uncertainties were assessed in rather different ways and ranged between about ±0.1 m per year and more than ±1.0 m per year (at 95% confidence intervals), often not overlapping with validation results or ensemble means.

The experiments also revealed quite obvious difficulties with each data source. While optical imagery has some limitations in
notoriously cloudy or textureless regions (the latter holding mostly for 8-bit sensors), data from radar missions have to cope with steep terrain during phase unwrapping as well as with seasonally and annually variable radar penetration. Looking in more detail at the different workflows, we found that DEM production, selection, and co-registration have the biggest impact on results. Compared to these three basic processing components, the other steps generally have a smaller impact but can become important for individual cases and depending on the use of optical or radar data.

Overall, we could not find if any single workflow performs best, as our experiment was limited to a small number of validation sites, but we identified several good practices on the impact of processing steps. Assessments of glacier elevation changes ideally have full control over the entire processing workflow and come with a validation using high-quality DEMs from airborne or higher-resolution satellite data. Here we see a great potential for a coordinated community effort to compile a benchmark dataset of validation DEMs for selected glaciers around the world. Future inter-comparison experiments could look
in more detail at the impact of different implementations of individual processing steps at the glacier scale or extend the scope to other sensors and entire glacier regions, in order to investigate whether the discrepancies found in this inter-comparison experiment for small areas cancel out over larger regions or remain. This could be achieved through community-driven software development that provides peer-reviewed, open-source, and transparent workflows on a public forum, such as GitHub. In addition, we see an urgent need to develop common good practices for uncertainty assessments and the material
density assignment that is required to convert elevation change to glacier mass balance. Altogether, these will improve the reliability and credibility of large-scale spaceborne glacier change assessments.



**Appendices**

**Appendix A**

This material complements the figures of the manuscript, which mainly are from Hintereis and/or Baltoro, with analogue ones
from all study sites (Figs. A1−A12).

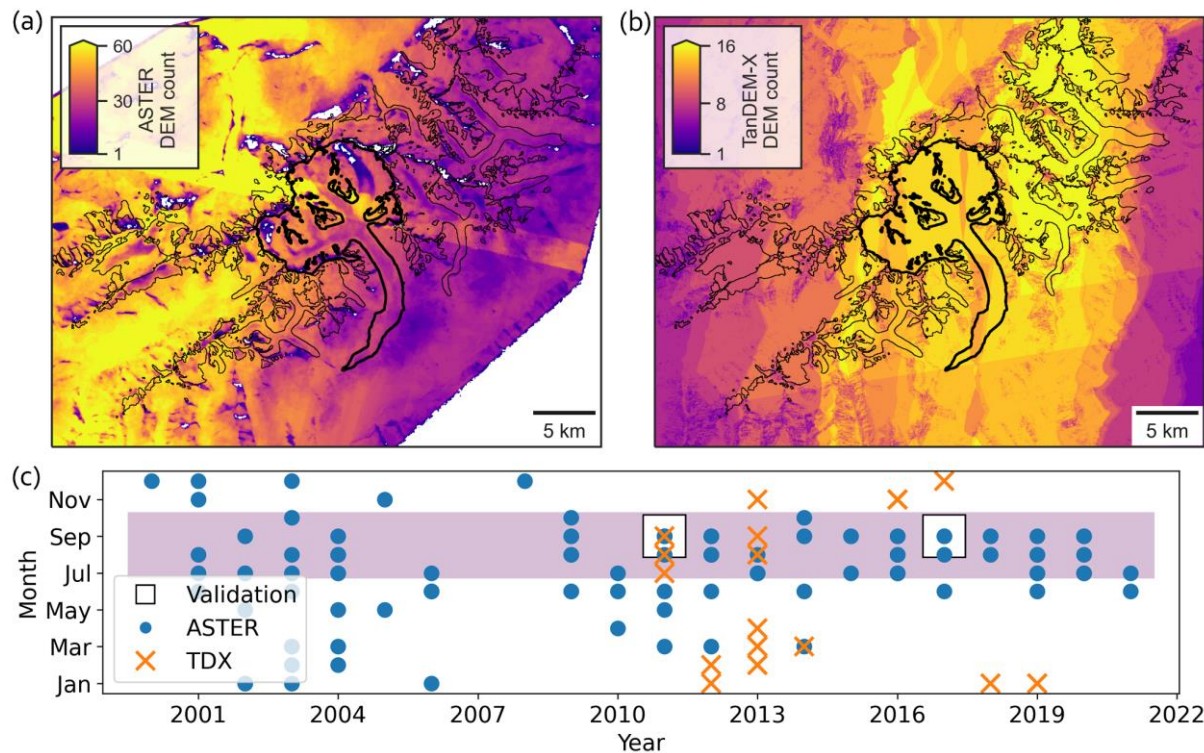

**Figure A1: Spatio-temporal coverage of the experiment DEMs over Aletsch (bold RGI 6.0 outline). The DEM count over the area is
shown for a) ASTER and b) TanDEM-X. Data voids in the DEMs are shown in white. c) Dates of the validation period and temporal
coverage of the TanDEM-X and ASTER DEMs provided for Aletsch. The summer months (July to October) are highlighted in
purple.**




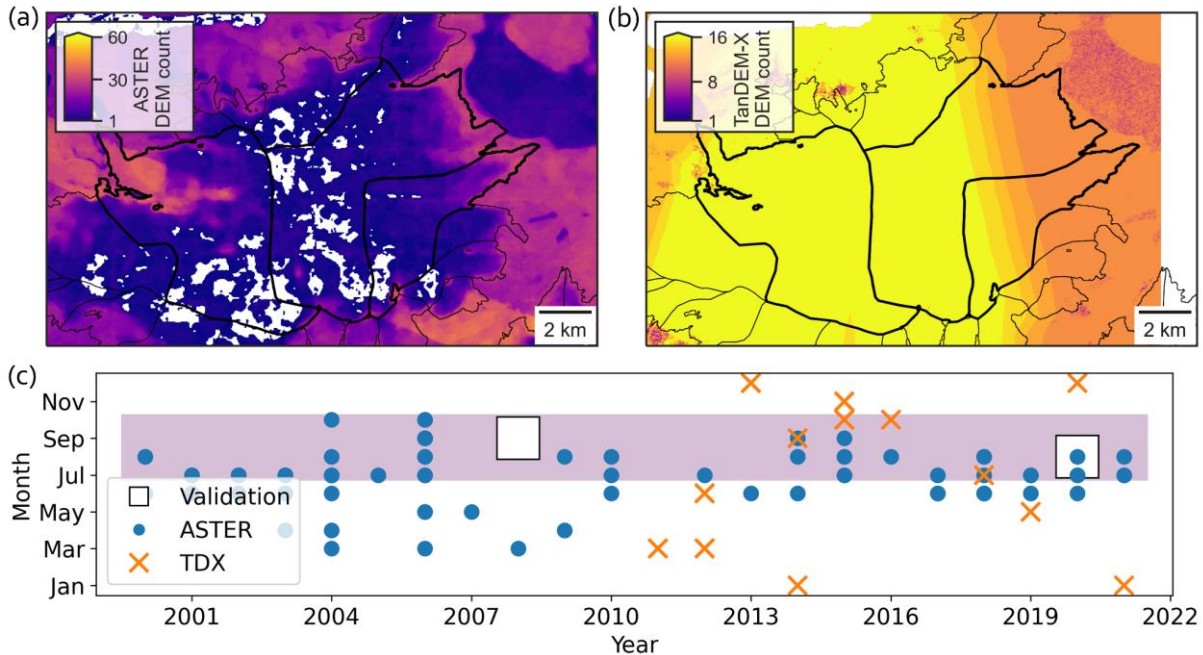

**Figure A2: Spatio-temporal coverage of the experiment DEMs over Vestisen (bold RGI 6.0 outline). The DEM count over the area is shown for a) ASTER and b) TanDEM-X. Data voids in the DEMs are shown in white. c) Dates of the validation period and temporal coverage of the TanDEM-X and ASTER DEMs provided for Vestisen. The summer months (July to October) are highlighted in purple.**

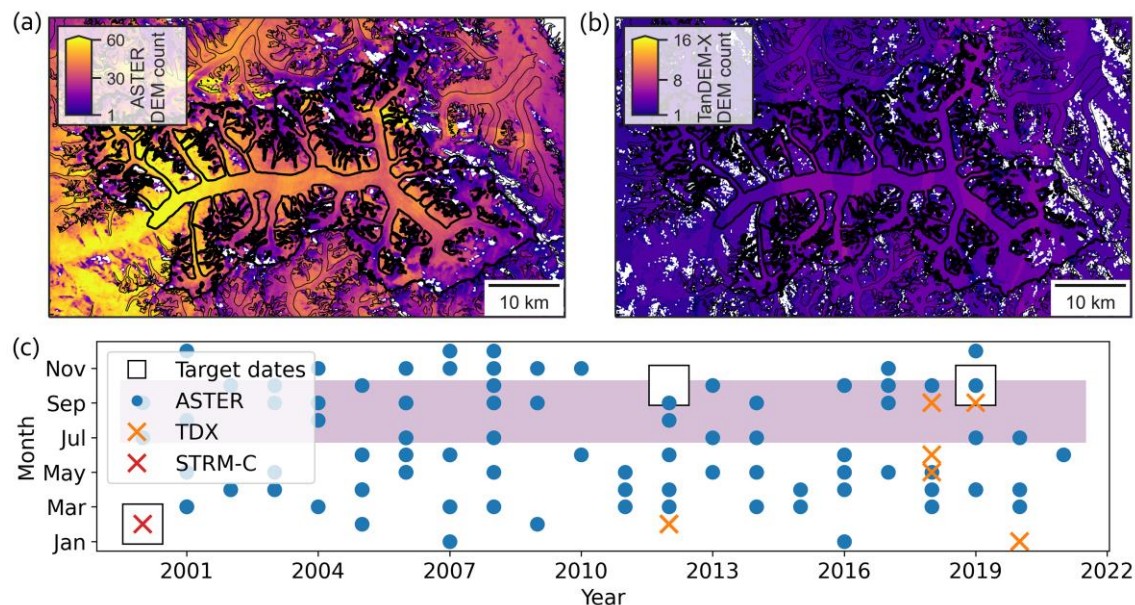

**Figure A3: Spatio-temporal coverage of the experiment DEMs over Baltoro (bold RGI 6.0 outline). The DEM count over the area is shown for a) ASTER and b) TanDEM-X. Data voids in the DEMs are shown in white. c) Dates of the validation period and temporal coverage of the TanDEM-X and ASTER DEMs provided for Baltoro. The summer months (July to October) are highlighted in purple.**





**Figure A4:** Spatio-temporal coverage of the experiment DEMs over Northern Patagonian Icefield (bold RGI 6.0 outline). The DEM count over the area is shown for a) ASTER and b) TanDEM-X. Data voids in the DEMs are shown in white. c) Dates of the validation period and temporal coverage of the TanDEM-X and ASTER DEMs provided for Northern Patagonian Icefield. The summer months (January to April) are highlighted in purple.





**Figure A5: Compilation of elevation change rate maps in metres per year for Aletsch. The spaceborne results, sorted by the type of sensor (optical and radar) and by DEM selection strategies (e.g., pair, mosaic, and time series approaches) are shown together with the airborne validation result. Group labels are represented by three-letter acronyms with corresponding result numbers (#). Results flagged as low confidence are indicated by group labels in *italic*. Glacier outlines are from RGI v6.0, showing the target and neighbouring glaciers with thick and thin outlines, respectively.**







**Figure A6: Compilation of elevation change rate maps in metres per year for Vestisen. The spaceborne results, sorted by the type of**
900 **sensor (optical and radar) and by DEM selection strategies (e.g., pair, mosaic, and time series approaches) are shown together with the airborne validation result. Group labels are represented by three-letter acronyms with corresponding result numbers (#). Results flagged as low confidence are indicated by group labels in *italic*. Glacier outlines are from RGI v6.0, showing the target and neighbouring glaciers with thick and thin outlines, respectively. Note that UIO-4 and UIO-5 do not show stable terrain as they used a time series approach based on elevation bins (cf. Table S12).**





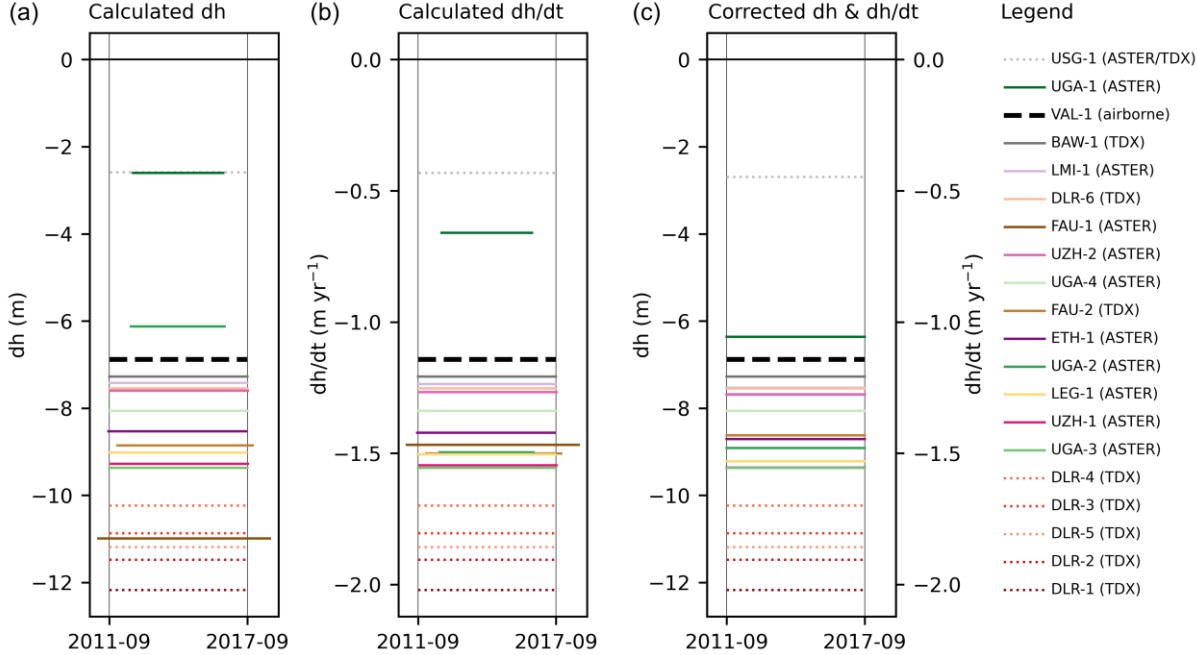

**Figure A7: Elevation changes of Aletsch from airborne validation and spaceborne results. The results are shown as originally calculated (a) mean elevation changes (dh) and (b) corresponding elevation change rates (dh/dt), as well as (c) after temporal corrections to the validation period, for contributions where this was not done by the participants. The spaceborne results are labelled for group and result number, with DEM sources in brackets. Low confidence results are marked by dotted lines. The airborne validation (VAL-1) result is shown as a dashed black line with the corresponding validation period marked by vertical grey lines. Error bars of the validation data are smaller than the line width. Error bars of the spaceborne results are shown in Fig. A11. The legend is sorted by descending elevation change (rate) values (c), which corresponds to negative and positive differences to the validation result.**



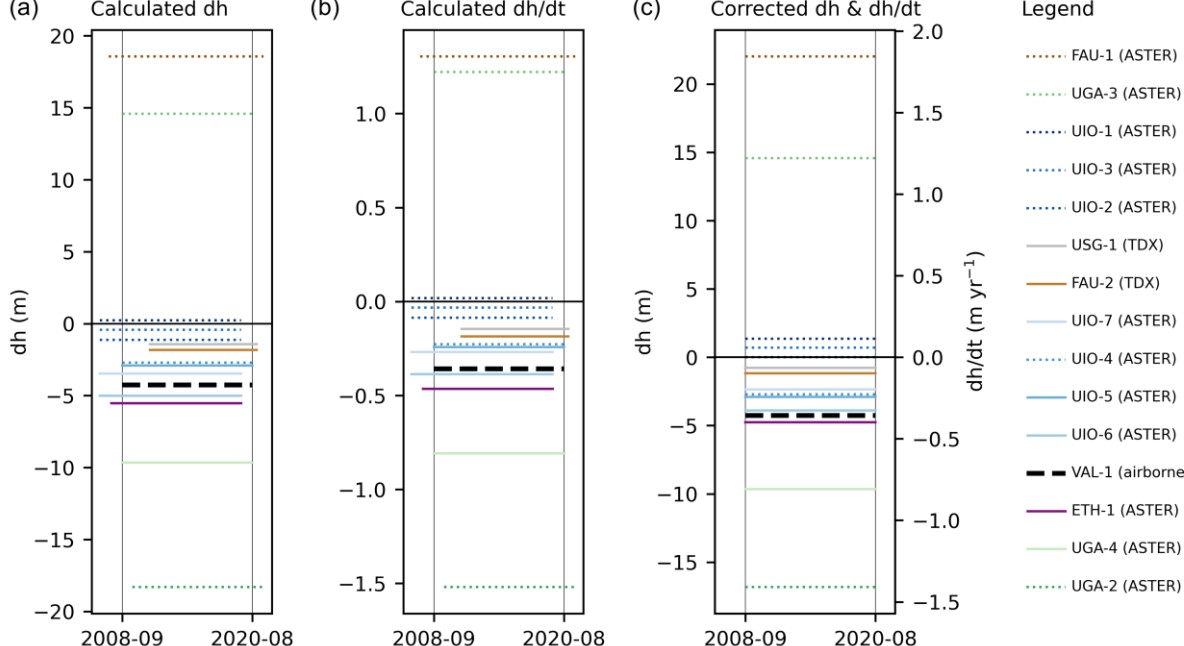

**Figure A8: Elevation changes of Vestisen from airborne validation and spaceborne results. The results are shown as originally calculated (a) mean elevation changes (dh) and (b) corresponding elevation change rates (dh/dt), as well as (c) after temporal corrections to the validation period, for contributions where this was not done by the participants. The spaceborne results are labelled for group and result number, with DEM sources in brackets. Low confidence results are marked by dotted lines. The airborne validation (VAL-1) result is shown as a dashed black line with the corresponding validation period marked by vertical grey lines. Error bars of the validation data are smaller than the line width. Error bars of the spaceborne results are shown in Fig. A12. The legend is sorted by descending elevation change (rate) values (c), which corresponds to negative and positive differences to the validation result.**





**Figure A9: Compilation of elevation change rates maps in metres per year for the Northern Patagonian Icefield for the target period 2000−2014 from the different DEM source combinations, i.e. SRTM/ASTER, ASTER/ASTER, and SRTM/TanDEM-X. Group labels are represented by three-letter acronyms with corresponding result numbers (#). Results flagged as low confidence are indicated by group labels in *italic*. Glacier outlines are from RGI v6.0, showing the target and neighbouring glaciers with thick and thin outlines, respectively. Note that a common spatial coverage by all groups could only be achieved for 55 glaciers on the eastern side of the icefield.**






**Figure A10: Compilation of elevation change rates maps in metres per year for the Northern Patagonian Icefield for the target period 2014−2019 from the different DEM source combinations, i.e. ASTER/TanDEM-X, ASTER/ASTER, and TanDEM-X/TanDEM-X. Group labels are represented by three-letter acronyms with corresponding result numbers (#). Results flagged as low confidence are indicated by group labels in *italic*. Glacier outlines are from RGI v6.0, showing the target and neighbouring glaciers with thick and thin outlines, respectively. Note that a common spatial coverage by all groups could only be achieved for 55 glaciers on the eastern side of the icefield.**






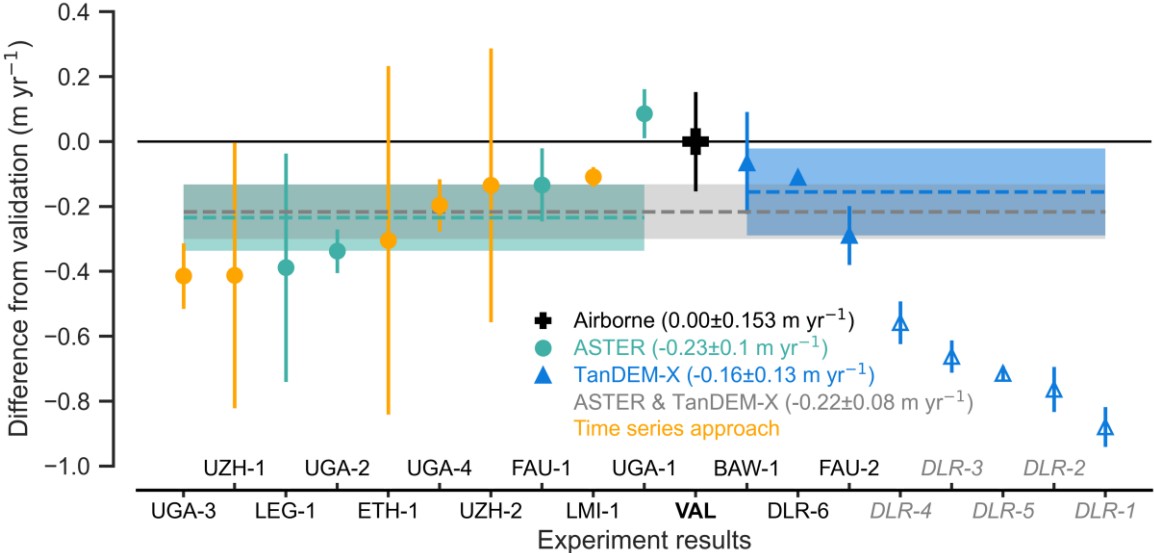

**Figure A11: Difference of the spaceborne results from the airborne validation data for Aletsch. Spaceborne results are shown**
**including temporal corrections (cf. Fig. A7c). Low confidence results are marked by empty symbols and by corresponding labels in**
*italic* **and grey. Results based on a time series approach are highlighted in orange. Reported uncertainties are shown as vertical lines.**
**Ensemble means (dashed lines) and related 1.96 standard errors (shadings) are shown, excluding low confidence results. Note that**
**the uncertainty of the validation result is smaller than the marker.**

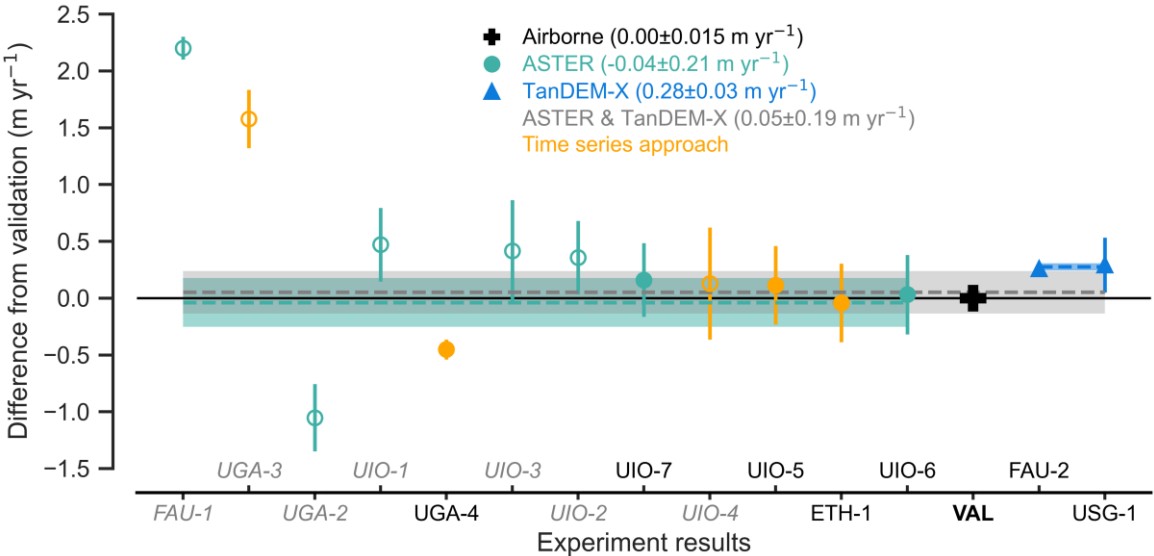

**Figure A12: Difference of the spaceborne results from the airborne validation data for Vestisen. Spaceborne results are shown**
**including temporal corrections (cf. Fig. A8c). Low confidence results are marked by empty symbols and by corresponding labels in**
**italic and grey. Results based on a time series approach are highlighted in orange. Reported uncertainties are shown as vertical lines.**
**Ensemble means (dashed lines) and related 1.96 standard errors (shadings) are shown, excluding low confidence results. Note that**
**the uncertainty of the validation result is smaller than the marker.**


**Code availability**

The code related to the implementation of the experiment is available from the participants partly on request and partly from public repositories as listed in Tables S1−S15.

**Data availability**

The data used in the present study can be accessed from the original data providers: Airborne validation data for Hintereis (2010: Prinz et al. (2023); 2019: RealityMaps (Digital Elevation Model of Hintereisferner based on airborne laser scanning survey of 21 September 2019. RealityMaps, Munich, Germany, 2023)), Aletsch (2011: Ginzler and Hobi (2015); 2017: Swisstopo (2023)), and Vestisen (2008 and 2020: NVE (2023)), SRTM DEM (USGS, 2017), Copernicus DEM (ESA and Airbus, 2022), glacier outlines (RGI Consortium, 2017), and mass balance observations (WGMS, 2021). ASTER L1A data
are available from EarthData portal (ASTER L1A Reconstructed Unprocessed Instrument Data, version 003. DOI: 10.5067/ASTER/AST_L1A.003, 2023) and the DEMs can be processed from the L1A granules with Ames Stereo Pipelines (DEMs processed from a batch of ASTER L1A images using Ames Stereo Pipelines (ASP), accessed from GitHub., 2023; Dussaillant et al., 2019). Co-registered Single look Slant range Complex (CoSSC) data of the TanDEM-X (Zink et al., 2012) mission is available from the German Aerospace Center (DLR) archive EOWEB GeoPortal (EOWEB GeoPortal, 2023a).

**Supplement link**

The supplement related to this article is available online at:

**Author contributions**

The first authors (M.Z., F.B., M.B.) designed the experiment. The first five authors (L.P., M.Z., C.S., F.B., M.B.) analysed the results and prepared the manuscript including figures and tables and supplemental materials. The following authors (listed in
alphabetical order) provided the experiment airborne validation data and spaceborne data (Supplement Tables S1−S15), participated in the experiment and commented on the manuscript. Co-contributors of validation and experiment data not involved in the manuscript preparation are acknowledged in Supplement Tables S1−S15.

**Competing interest**

At least one of the (co-)authors is a member of the editorial board of The Cryosphere.



## Acknowledgements

The present work is a contribution to the working group on Regional Assessment of Glacier Mass Change (RAGMAC) of the International Association of Cryospheric Sciences (IACS) and to the Glacier Mass Balance Intercomparison Exercise (GlaMBIE) of the European Space Agency (ESA). We thank T. Barnhart (USGS) and K. Arie (JAXA) for their feedback on the manuscript. The study was carried out as a community effort with partial support from IACS and from the following projects of participants: L.A.: "Regional massebalanseestimat av norske breer" (#N80524); E.B.: French Space Agency (CNES); C.F., C.M., L.S.: U.S. Geological Survey Ecosystems Climate R&D Program, Benchmark Glacier Project (USG); R.H.: University of Toulouse; R.H., M.H.: Swiss National Science Foundation (#184634); L.P., D.T, A.K.: ESA project Glaciers_cci+ (#4000127593/19/I-NB); D.T.: Research Council of Norway projects MASSIVE (#315971) and SNOWDEPTH (#325519); A.K.: ESA EE10 Harmony (4000135083/21/NL/FF/ab). M.Z.: Federal Office of Meteorology and Climatology MeteoSwiss within the framework of GCOS Switzerland. M.B., C.S.: Deutsche Forschungsgemeinschaft (PN 313916628). Any use of trade, firm, or product names is for descriptive purposes only and does not imply endorsement by the U.S. Government, or of any other institution.

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
