# Peer review of "Observing glacier elevation changes from spaceborne optical and radar sensors – an inter-comparison experiment using ASTER and TanDEM-X data"

_EGUsphere, 2023_

## Referee Comment (RC2)

The study outlines the outcomes of an inter-comparison experiment aimed at estimating glacier elevation changes using spaceborne optical stereo (ASTER) and synthetic aperture radar interferometry (TanDEM-X) data. The study also emphasizes the importance of accurate glacier mass change observations for understanding climate change impacts, such as regional runoff, ecosystem changes, and global sea-level rise.

I have some general comments to further improve this manuscript. Please specify the range of results obtained from the various studies, indicating both the differences and similarities. Quantifying these differences can provide a clearer picture of the challenges in current methodologies. Clearly articulate the significance of the findings in terms of their implications for understanding climate change impacts, emphasizing how the observed discrepancies impact broader applications such as regional runoff, ecosystem changes, and sea-level rise. Also, mention in one line about the details on the validation process with airborne data, including specific metrics used and the degree of accuracy achieved. Specify the level of community involvement in the experiment and how collaborative efforts contributed to the study. This can highlight the strength of community-based research. Encourage the adoption of transparent practices within the scientific community. Provide a concise summary of the major findings and contributions at the beginning or end of the abstract for better reader orientation.

- Elaborate on the methodologies used for bias corrections, co-registration, outlier filtering, void filling, radar signal penetration, and temporal corrections.
- Consider including a flowchart or a schematic representation of the data processing steps to enhance the clarity of the methodology section.
- Expand on the comparative analysis between optical and radar data, addressing the limitations and methodological differences. This will provide a more nuanced understanding of the challenges and strengths associated with each data type.

I believe by incorporating these suggestions and comments will refine the manuscript, making it clearer on the study's findings, their importance, and how collaboration shaped the research, thus making it suitable for publication.

Specific comments for each section are.

**Introduction**

- I will be better to begin by clearly stating the motivation behind the study and the specific research gap it aims to address. This could involve highlighting the importance of accurately assessing glacier mass balance for understanding climate change impacts and informing adaptation strategies.

- Provide a more detailed overview of the existing methods and techniques used to assess glacier elevation changes. This could involve discussing traditional mapping techniques, topographic surveys, and the role of digital elevation models (DEMs) derived from different sensor platforms.

- Emphasize the significance of spaceborne optical and radar sensors in revolutionizing glacier monitoring efforts. Discuss the advantages and limitations of optical and radar data for observing glacier elevation changes, including their spatial and temporal coverage, resolution, and sensitivity to environmental conditions.

- Introduce the ASTER and TanDEM-X satellite missions more prominently, including key characteristics such as sensor type, temporal coverage, and data processing capabilities. Briefly discuss the evolution of these missions over time and their contributions to glacier research.

- Clearly outline the objectives and methodology of the inter-comparison experiment described in the paper. Explain why comparing results from ASTER and TanDEM-X data is important, and how this contributes to improving our understanding of glacier elevation changes.

- Add some more relevant references such as Kääb et al., 2012; Bolch et al., 2011; Scherler et al., 2011; IPCC. (2021)

**Data section**

Ensure that the description of each component of the data is clear and concise. Provide sufficient details to allow readers to understand the study sites, airborne validation data, DEMs (ASTER, TanDEM-X, SRTM), and auxiliary data without overwhelming them with unnecessary information.

- Define any acronyms or specialized terms used in the description of the data to avoid confusion.
- Offer some context for why each type of data was selected for the study. Explain the rationale behind choosing specific study sites, DEMs, and auxiliary data, highlighting their relevance to the research objectives.
- If there are any unique aspects or novel approaches in the data collection or processing, highlight these to emphasize the original contributions of the study.
- Ensure that proper citations are provided for any previously published data sets or auxiliary data used in the study. This helps readers to trace the origins of the data and provides credit to the original sources.

**Study Area**

- Explain in more detail the criteria used for selecting the study sites, such as glacier size, topography, location, and availability of validation data. This will help readers understand why these specific sites were chosen for the inter-comparison experiments.
- Clearly articulate how each study site poses various data processing challenges for both optical and radar sensors. Provide specific examples of these challenges and explain their implications for DEM differencing.
- Where possible, provide supporting evidence or references to previous studies that have documented the challenges faced at each study site.
- Emphasize any unique characteristics or features of each study site that make it particularly interesting or challenging for the inter-comparison experiments. This could include factors such as glacier morphology, climate conditions, or geographical location.

**Glacier outlines**

- Clearly state the purpose of using glacier outlines from the Randolph Glacier Inventory (RGI) version 6.0 in the study. Explain how these outlines serve as a common reference

for comparison purposes and facilitate standardized analyses across different study sites.

- Acknowledge the limitations of using fixed glacier outlines, particularly in regions where glacier area changes significantly over time. Explain how this may introduce bias into the calculated specific elevation changes, especially in dynamic glacier environments such as the Alps.

- Maintain consistency in naming and abbreviations when referring to the RGI and its version number throughout the section.

**Airborne validation DEMs**

- Clarify why validation data from airborne lidar and aerial stereo images are important for the study. Explain how these data serve as ground truth measurements to assess the accuracy of spaceborne DEMs and validate their performance.

- Provide more detailed information about the acquisition dates, resolution, and sources of the airborne lidar and aerial stereo DEMs for each study site. This will help readers understand the characteristics and quality of the validation data used in the analysis.

- Describe the steps taken to process and prepare the airborne validation DEMs for comparison with spaceborne DEMs. This could include details on resampling, co-registration, and quality control procedures to ensure consistency and accuracy in the validation data.

- Discuss any limitations or challenges associated with the airborne validation DEMs, such as data gaps, varying acquisition dates, or potential artifacts. Address how these factors may impact the interpretation of validation results and the overall accuracy assessment.

- Include information on the accuracy of the validation data, particularly regarding the accuracy of elevation change measurements. Discuss how uncertainty was quantified and its implications for the interpretation of validation results.

- Acknowledge the absence of validation data for Baltoro and Northern Patagonian Icefield and discuss any potential implications for the study. Consider suggesting future research directions or alternative approaches for validating DEMs in these regions.

**2.5 Spaceborne experiment DEMs**

**2.5.1 ASTER DEMs**

- Clarify the significance of using ASTER DEMs for the study and explain how they contribute to the overall objectives. Discuss why ASTER data were chosen, emphasizing their global coverage and multi-temporal nature.

- Describe the process of acquiring and processing ASTER L1A images to generate DEMs for each study site. Provide information on the specific algorithms and parameters used for DEM generation, including any preprocessing steps or data projections applied.

- Discuss the quality of ASTER DEMs, including reported precision and potential limitations such as cloud cover, sensor saturation, and acquisition footprint. Address how these factors may affect the accuracy and spatial coverage of the DEMs, particularly in glacierized areas.

- Quantify the spatial coverage of ASTER DEMs for each study site, including the number of available DEMs and the variation in data points per glacier pixel. This will provide readers with a clear understanding of the spatial distribution and density of DEM data.
- Discuss the main challenges associated with using ASTER DEMs, such as incomplete spatial coverage and voids in the accumulation area. Provide insights into potential strategies or techniques used to address these challenges, such as interpolation or filtering methods.

**2.5.2 TanDEM-X DEMs**

- Clarify the significance of using TanDEM-X DEMs for the study and explain how they contribute to the overall objectives. Discuss why TanDEM-X data were chosen, emphasizing their high resolution and near-complete coverage of the glacier areas.
- Describe the process of acquiring and processing TanDEM-X data to generate DEMs for each study site. Provide information on the interferometric workflow, including steps for concatenating overlapping scenes, creating interferograms, and unwrapping phase data.
- Discuss the reported precision of TanDEM-X DEMs and any limitations or potential sources of error associated with their use. Address how accuracy may vary in mountainous terrain and how errors are mitigated during data processing.
- Quantify the spatial coverage of TanDEM-X DEMs for each study site, including the percentage of coverage on the glacier and the presence of voids or masked areas off-glacier. This will provide readers with a clear understanding of data availability and potential limitations.
- Include visual aids, such as figures or maps, to illustrate the spatial distribution of TanDEM-X DEMs over the study sites. This can help readers visualize the extent of data coverage and identify areas with potential data gaps or errors. Discuss the temporal coverage of TanDEM-X DEMs and how it varies between study sites due to the acquisition strategy of the mission. Address any challenges or limitations associated with the campaign-based acquisition mode and variable acquisition months.

**3 Methods**

**3.1 Inter-comparison experiment description**

- Describe in more detail the different strategies, post-processing steps, and corrections applied by participants to calculate their spaceborne estimates. This could include specific methods used for DEM selection, filtering, co-registration, and uncertainty estimation.
- Provide clearer explanations of the two experiments conducted as part of the inter-comparison study. Clearly state the objectives, target periods, and study sites involved in each experiment. This will help readers understand the specific goals and scope of each experiment.
- Discuss the rationale and methodology behind the temporal corrections applied in the first experiment, particularly for aligning spaceborne estimates with airborne validation

data. Explain how temporal discrepancies were addressed and why they are important for the accuracy of elevation-change estimates.

- Provide more details about the sensitivity study conducted in the second experiment, including the specific processing steps evaluated and their potential impact on glacier elevation change estimates. Discuss the significance of this sensitivity analysis for understanding the robustness of results.

**3.2 Participants and spaceborne results**

- Discuss the variability in the number of submissions for each study site. Explore potential reasons why some sites received more submissions than others, such as site characteristics, data availability, or participant preferences.
- Provide more detail about the different DEM sources and approaches employed by participants for each study site. Discuss any notable differences or trends in the types of data sources and methodologies used.
- Highlight the prevalence of ASTER DEMs over TanDEM-X DEMs in the submissions. Discuss potential reasons for this discrepancy and any implications it may have for the interpretation of results.
- Provide context for why Hintereis and Baltoro were chosen to illustrate the experiments in the main body of the manuscript. Discuss any unique characteristics or significance of these sites compared to the other study sites.

**3 General workflow**

- Expand on the description of each step in the workflow to provide more detail about the specific processes involved. For example, explain the methods used for bias correction, co-registration, noise filtering, and void-filling, and discuss the rationale behind each process.
- Highlight any key differences in approach or methodology used by different groups.
- Provide examples or case studies to illustrate how the workflow steps were applied in practice for specific study sites.
- Discuss the various types of corrections that can be applied in the workflow, such as temporal corrections to match the target period or corrections for radar signal penetration. Explain the purpose of each correction and how it contributes to improving the accuracy of the elevation change estimates.
- Encourage readers to refer to supplementary tables for more detailed descriptions of the DEM selection and processing strategies used by each group for each site. Provide a brief overview of the information available in these tables and its relevance to understanding the workflow.

**3.3.2 Spatial bias correction**

- Provide a clearer definition and explanation of the spatial trends and vertical deformation mentioned. Explain how these biases occur due to sensor behaviour, data acquisition, and processing. Specify the magnitude and extent of the biases more precisely, including the range of horizontal scales affect. Clarify the concept of "height

of ambiguity" and its role in generating the observed linear trends in TanDEM-X DEMs..

- Provide more detail on how each correction technique works and why it is effective. Explain the rationale behind fitting planes, polynomial functions, or sine functions, and how they address specific biases.

- Include references to relevant literature to support the choice of correction techniques and their effectiveness. Provide proper attribution for the correction techniques mentioned, citing the original sources or key references where these methods were introduced or validated.

- Discuss any quantitative assessments or validations of the correction techniques used. Provide details on how the effectiveness of each technique was evaluated and any metrics used to measure improvement in DEM quality. include statistical analysis or comparisons to demonstrate the impact of the corrections on reducing biases. Discuss the broader context of spatial bias correction in DEMs, including common challenges, emerging trends, and areas for future research.

**3.3.3 DEM co-registration**

- Start by clearly stating the importance of DEM co-registration in minimizing systematic errors before performing DEM differencing. Explain how shifts and rotations resulting from georeferencing techniques and processing distortion can impact the accuracy of DEMs.

- Discuss the variability in co-registration approaches adopted by different groups in the experiment. Explain the reasons behind the selection of specific stable terrain areas and reference DEMs.

- Describe the co-registration algorithms used by the groups, including specific references to equations or methodologies.Discuss the strengths and limitations of each algorithm in addressing shifts, rotations, and scaling between DEMs.

- Describe how remaining biases after co-registration were addressed, including any criteria used to determine whether biases were significant enough to warrant correction.

**3.3.4 Noise filtering and void-filling**

- Start by providing a brief overview of the types of artefacts and noise commonly found in ASTER and TanDEM-X DEMs, along with their causes. This will help readers understand the significance of noise filtering and void-filling.

- Clearly explain the sources of artefacts in ASTER and TanDEM-X DEMs, including sensor characteristics, surface properties, and topographic factors. Provide examples of artefacts such as sinks, phase unwrapping errors, shadowing, and layover.

- Clearly explain the sources of artefacts in ASTER and TanDEM-X DEMs, including sensor characteristics, surface properties, and topographic factors. Provide examples of artefacts such as sinks, phase unwrapping errors, shadowing, and layover.

- Provide a detailed explanation of the noise filtering approaches adopted by the participants, including the use of statistical parameters like standard deviation, NMAD, and threshold values. Clarify how these approaches help remove gross errors and artefacts from the DEMs.

- Highlight any novel or innovative approaches used by certain groups for noise filtering and void-filling. Discuss the advantages and limitations of these approaches compared

to more traditional methods. Ensure that relevant references and citations are provided to support the discussion of noise filtering and void-filling techniques.

**3.3.5 Radar penetration correction**

- Clarify the significance and implications of radar penetration correction in the context of glacier studies. Elaborate on the methods employed by different groups to address radar penetration. Provide a clearer explanation of how radar penetration correction was implemented in the study.

- Explain how inaccurate correction can lead to biased results and affect the reliability of glacier mass balance assessments. Emphasize the role of penetration depth variability in influencing the accuracy. Provide a detailed explanation of the probabilistic framework employed by the UST group and how it accounts for elevation-dependent penetration using a Gaussian probability distribution. Similarly, explain the specifics of the elevation-dependent C-band penetration model applied by the GAC group, referencing the findings of Kumar et al. 2019 of InSAR measurements.

**3.3.6 Temporal corrections**

- Begin by clearly stating why temporal corrections are necessary when comparing different datasets, especially when spaceborne observation dates differ from the airborne validation period.

- Provide a more detailed explanation of the various strategies employed by participants for temporal corrections. Describe each strategy (no temporal correction, linear scaling, annual corrections using glaciological observations, non-linear regressions) and clarify the assumptions underlying each approach.

- Discuss the significance of the remaining temporal differences after applying corrections. Explain why these differences occurred despite the correction efforts and discuss their potential impact on the accuracy of the elevation change estimates. This discussion will provide insights into the limitations of the correction methods and the challenges involved in achieving precise temporal alignment.

- Provide a clear explanation of the approach used by Zemp and Welty (2023) to correct for the remaining temporal differences. Describe how seasonal observations of glaciological mass balance were temporally downscaled using sine functions and how this adjustment was applied to spaceborne elevation change results.

- Quantify the effects of the corrections on the temporal differences between spaceborne elevation changes and validation dates. Provide specific examples or statistics to illustrate the magnitude of the corrections and their impact on the elevation change estimates.

**3.4 Uncertainty assessment**

- Provide a clearer explanation of how the overall uncertainty was calculated by the participants. Describe the process of quadratic summation of different error sources and how the 95% confidence interval was derived.

- Expand on the different error sources considered by participants and the methods employed to quantify them. Provide additional context and explanation for each error source, including pixel elevation change error, errors in glacier outline and area mapping, errors due to missing observations, and errors related to temporal mismatch or radar penetration correction. This will help readers understand the specific challenges addressed by each group and the methodologies used to mitigate them.
- Discuss the variation in error quantification methods employed by different groups. Highlight any differences in approaches, assumptions, or model parameters used for error estimation.

**4 Results**

- Provide additional clarification on why participants preferred using the provided DEMs rather than self-processed DEMs. Elaborate on any advantages or limitations of using provided DEMs and discuss how this choice may have influenced the results.
- Expand on the correction of radar signal penetration by two groups for the Baltoro site and the absence of such correction for the Northern Patagonian Icefield. Discuss the implications of this difference in correction approaches for the accuracy and reliability of the results. Additionally, provide context on the challenges or limitations of radar signal penetration correction and how participants addressed them.
- Discuss the limited application of temporal corrections by participants and the relatively low participation in sensitivity studies. Provide insights into the reasons for these trends and discuss their implications for the reliability and robustness of the results. Additionally, highlight any key findings or observations from the sensitivity studies that were conducted.
- Provide clarity on the availability of spaceborne results and related processing steps for each study site. Ensure that readers understand where they can access this information and how it can contribute to their understanding of the study findings. Consider providing direct links or references to the tables containing this information for each site.

**4.2 Elevation change assessment for glaciers with airborne validation data**

- Provide a brief overview or introduction to the first experiment, outlining its objectives and the glaciers studied (Hintereis, Aletsch, and Vestisen). This will help orient readers and provide context for the subsequent discussion of the results.
- Offer a more detailed explanation of the observed elevation change patterns for each glacier. Describe the main findings regarding glacier-wide mean elevation changes, trends in ice loss, changes in accumulation areas, and any notable spatial variations. Providing this information will help readers understand the key findings and interpretations derived from the comparison of spaceborne results with airborne validation data.
- Discuss the accuracy and reliability of the spaceborne results compared to the airborne validation data. Address any discrepancies or differences observed between the two datasets, including variations in spatial coverage, noise levels, and patterns of elevation change. Consider discussing potential sources of error or uncertainty in the spaceborne results and their implications for the interpretation of the findings.

- Discuss any insights gained from comparing spaceborne results with airborne validation data and highlight areas for future research or improvement in methodology. This will provide closure to the discussion and guide readers on the significance of the results presented.

**4.3 Regional-scale elevation-change assessment and sensitivity study**

- Provide a brief introduction or context for the second experiment, outlining its objectives and focus on regional-scale elevation change assessment for Baltoro and the Northern Patagonian Icefield.
- Clarify the main focus of the second experiment, which was to analyse the effect of various processing chain steps rather than comparing spaceborne results with airborne validation data. Explain why this approach was chosen and how it contributes to understanding the sensitivity of elevation change estimates to different processing methods.
- Offer a detailed description of the observed elevation change patterns for both Baltoro and the Northern Patagonian Icefield. Discuss any notable features or trends identified in the elevation change rate maps, such as surge-type patterns, spatial variability, and differences between observation periods. Providing this information will help readers visualize the results and understand the variability in elevation change estimates.
- Discuss the spread of calculated elevation differences and mean elevation change rates for both study regions and observation periods. Explain the significance of the variability in results and any trends observed between the two periods. Address any factors contributing to the spread in results, such as differences in input data sources and processing techniques.

**Discussion**

- Address any additional considerations or issues that were not covered in the experiments but may have a significant impact on geodetic glacier mass change assessments, such as changes in glacier area and density conversion factors. Discuss the implications of these factors for interpreting elevation change estimates and comparing them to other sources of data.
- Provide clear and concise explanations of the findings presented in the results section, highlighting the main conclusions drawn from the analysis. Use specific examples and references to support your arguments and illustrate key points.
- Offer thoughtful interpretations of the findings and discuss their implications for glacier elevation change assessment using spaceborne data. Consider addressing questions such as the reliability of spaceborne estimates, the significance of systematic differences between approaches, and the potential impact of processing steps on the accuracy of elevation change estimates.
- Compare and evaluate the different processing steps and approaches used by the participating groups in the experiments. Discuss the strengths and limitations of each approach, as well as any observed trends or patterns in the results.
- Address any additional considerations or issues that were not covered in the experiments but may have a significant impact on geodetic glacier mass change assessments, such as changes in glacier area and density conversion factors. Discuss

the implications of these factors for interpreting elevation change estimates and comparing them to other sources of data.

- Highlight areas for future research and methodological development based on the gaps and limitations identified in the experiments. Discuss potential avenues for further investigation, such as the development of new processing techniques, the integration of additional data sources, and the exploration of advanced uncertainty quantification methods.
- By summarizing the main insights gained from the inter-comparison experiments and discussing their broader implications for the field of glacier remote sensing. Offer final reflections on the significance of the findings and their relevance for advancing our understanding of glacier dynamics and climate change.

**Conclusion**

- Begin the conclusions section by summarizing the main findings of the study, focusing on aspects such as the large spread in elevation change estimates, the challenges associated with individual results from spaceborne surveys, and the potential of ensemble approaches to reduce random errors.
- Discuss the challenges and limitations associated with uncertainty assessment and validation of elevation change estimates. Address the variability in reported uncertainties and the importance of using high-quality DEMs for validation purposes. Highlight the need for a coordinated community effort to compile a benchmark dataset of validation DEMs for selected glaciers worldwide.
- Provide recommendations for future research based on the findings of the study. Discuss potential avenues for further investigation, such as conducting more extensive inter-comparison experiments, exploring the impact of different processing steps at the glacier scale, and developing common good practices for uncertainty assessment and material density assignment.

---

## Author Comment (AC1)

Title: **Observing glacier elevation changes from spaceborne optical and radar sensors – an inter-comparison experiment using ASTER and TanDEM-X data**

Author(s): Livia Piermattei et al.
MS No.: egusphere-2023-2309
MS type: Research article

Handling editor: Vishnu Nandan

**Author's response related to the comments received to the manuscript:**
https://doi.org/10.5194/egusphere-2023-2309

We thank the two reviewers and the editor for their feedback on our manuscript. We revised our manuscript under consideration of their comments and suggestions, which helped to improve the clarity of our paper.

The main changes we made to the manuscript are as follows:
- We improved the method and discussion sections by clarifying the challenges for optical and radar data and the solutions implemented by the participants within their processing chain.
- Throughout the manuscript, we improved the description and clarity of the challenges of radar data, especially in mountainous areas, and the impact of radar penetration on the assessment of elevation changes.
- We adjusted Figure 4 by adding a list of all participants for each group.

We are convinced that these revisions - as detailed in the point-by-point reply below - addressed all the comments raised by reviewer 1 and the editor (based on the feedback from reviewer 2) and made our manuscript more understandable.

**Point-by-point reply to reviewer comments**: editor and reviewer comments are in black, and the authors' response in blue, with citations from the revised manuscript in green.

……………………………

**RC2**: Editor, 21 Feb 2024

We note that the anonymous reviewer 2 provided an extensive report, which lists more than 90 generically formulated points, each with a list of sub-points. Unfortunately, these points are too generic to be addressed effectively, and many of these points have already been implemented in the manuscript or are clearly beyond the scope of the study. In fact, this report reads to us as if it was produced using Artificial Intelligence assistance, which would be against EGU recommendations: https://www.egu.eu/news/1031/statement-on-the-use-of-ai-based-tools-for-the-presentation-and-publication-of-research-results-in-earth-planetary-and-space-science/

Consequently, in agreement with the Editor, we have limited the response to the points below, which are selected and further clarified by the Editor, based on the original comments from reviewer 2.

**Specific comments:**

Line No. 318: While the authors have indicated that details of corrections applied by each group are provided in tables S4 to S15, it would be beneficial to include a note on the lack of

incorporation of seasonality corrections by many groups. Additionally, specify the buffer area chosen by each group to consider the stable area?

Response: We agree that most participants did not apply seasonal correction except for two groups (ETH and LMI). However, this depends also on the type of approaches used (e.g., pair or time series) and the selected DEMs. For example, in the case of a pair approach, where the two DEMs are very close in date to the target period but with different years, an annual correction was required rather than a seasonal correction. This was the case for the ASTER pair approach in the Hintereisferner case study. In the discussion section (lines 714-734), we explained the type and impact of the temporal correction according to the survey date.
However, in subsection 3.3.6 about temporal correction, we added a sentence summarising the main corrections used by the participants.
Line 452: "It is important to note that most groups did not apply seasonal corrections. However, the type of corrections applied depended on the approach used to estimate the elevation change (i.e., single pair or time series and related regression approaches) as well as the closeness of the selected DEMs to the target period when using the pair approach."

Thank you for asking about the buffer. Indeed, we mistakenly reported a buffer of the glacier outline in the DEM co-registration section, while participants only applied the buffer for the area uncertainty calculation in section 3.4. We removed the sentence about the RGI buffer (line 374).

Line no 399-401, 403, 409-410: Indicate theoretical penetration depths that has been observed from these studies, with respect to C- and X-band wavelengths. The authors do provide a good literature reference but could be improved if you could provide a quantified threshold to the depths observed on both cold and warm glacier ice and through 'snow' (tricky right?).

Response: As mentioned in the discussion section (Line 754), radar penetration is challenging and requires further research to determine its impact in different areas of interest and for different wavelengths. Setting a threshold for cold and temperate ice is difficult, as this will strongly depend on snow depth and underlying firn density/structure - and, therefore, on the climate and the relative elevation of the glacier.
According to our experiment, we observed a maximum penetration of up to 8 meters on the accumulation area of Aletsch glacier, a finding confirmed by other studies such as Leinss and Bernhard (2021) and Bannwart et al. (2023) over the same glacier. As written below, we reported the penetration depth estimates of C- and L-band from the studies by Dall et al. (2001) and Rignot et al. (2001).
Line 423: "... For TanDEM-X, the penetration depth is within a few metres, but average penetration depths of up to several metres (> 5 m) have been reported previously on Aletsch Glacier (Leinss et al., 2021; Bannwart et al., 2023) and other mountain glaciers in the Alps and High mountain Asia (Dehecq et al., 2016; Li et al., 2021) as well as the ice sheets (Abdullahi et al., 2018; Rott et al., 2021), while for the longer wavelength C-band of SRTM, even higher penetration depths can occur (Dall et al., 2001; Rignot et al., 2001). Dall et al. (2001) show that the height difference between lasers and InSAR in East Greenland changes from 0 m to a maximum of 13 m in the soaked and percolation zones, respectively. Similarly, Rignot et al. (2001) discovered that the radar penetration depth of C- and L-bands varies in different zones of Greenland (e.g., cold polar firn, exposed ice surface and marginal ice) and ranges from 1 to 15 m. For temperate glaciers in Alaska, the penetrations are between 4 and 12 m in C- and L-bands, respectively, with little dependence on snow/ice conditions. These studies highlight the challenge of establishing a radar penetration threshold for cold and temperate ice, as it depends on snow depth, underlying firn density/structure, climate, and the relative elevation of the glacier."

Figure 4 and section 3.x.x.x. – I wonder if there is a better way to show all 12 groups in the figure and in text. Readers have to refer to Supplement to know more about the groups and the members, which sometimes can be annoying. Can you provide a table in the manuscript referring to these members and groups. I know it will be cluttered but atleast useful not to check multiple documents at the same time. Think about it!

Response: We think that the Sankey diagram clearly shows how the study sites and DEM sources are connected to different groups. However, we agree that making the group members more visible is important. To address this, we included a table in Figure 4 that lists the names of the participants in each group with their main institution. Thank you for this suggestion.

[Figure]

"Figure 4: … Sankey produced with Sankeymatic (SankeyMatic, 2023). At the bottom is the list of the participants in each group with their main institution."

The explanation for UIO-4 and UIO-5 not showing stable terrain due to their use of a time series approach based on elevation bins (Table S12) is somewhat vague. Please provide a more detailed explanation of why the time series approach they used could provide stable area changes?

Response: Thank you. We provided more details in the figure caption about the lack of stable area in the dh/dt map.
Line 531: "Note that UIO-3 and UIO-4 do not show stable terrain as they used a time series approach based on elevation bins within the glacier area (cf. Table S12)."

Line No. 684-685: The authors mentioned, "Different tools, in combination with a priori or posteriori bias corrections, were used to correct shifts, rotations, and scale effects between DEMs." To enhance clarity, elaborate on the specific bias correction methods used instead of referring to them as a priori or posteriori? Additionally, instead of writing "a few meters," specify the value of the correction factor applied for the study?

Response: Thank you for the suggestion. We agree that the sentence regarding bias correction was unclear. We have now clarified it and explained what "corresponding corrections" means. Instead of repeating the methods used by the participants, we prefer to refer to the method section regarding the co-registration and bias approaches. Additionally, we replaced the generic phrase "few metres" with a specific value, as reported below.
Line 713: "... Different co-registration tools (Sect. 3.3.3) were used to correct shifts, rotations, and scale effects between DEMs in combination with bias corrections applied before or after the co-registration to correct spatial trends and vertical deformation (Sect. 3.3.2). Figure 10 illustrates that co-registration corrections can be up to 2 metres per year and, hence, …."

Line 704: The authors stated, "Participants have applied various void-filling techniques to calculate glacier-wide elevation changes." To improve understanding, provide examples of the techniques applied for void filling?

Response: Thank you. We prefer not to report additional details in the discussion section. Our aim here is to summarise the main factors that impact elevation change estimates and highlight which factor has the most impact according to different approaches and data. However, we added the reference to the method section where we described the solutions implemented by the participants for the void filling (line 392). In the method section, we rephrased the sentence and provided more details about the used approaches.
Line 408: "Voids in the DEM difference maps were filled by almost all groups using spatial methods such as the local or global hypsometric approach (McNabb et al., 2019). This method divides the glacier into elevation bins ranging from 20 m to 100 m intervals and assigns the average elevation change of the corresponding elevation bin to the data voids. Additional solutions included the weighted version of the local hypsometric method (ETH, Hugonnet et al., 2021), and the UIO group also applied a simple Inverse Distance Weighting interpolation."

Line No. 765-774: Regarding glacier area, did the authors compare RGI 7 glacier data with other globally available glacier boundaries for Baltoro Glacier (Karakoram, Pakistan) and other data for High Mountain Asia, such as GAMDAM? Clarify

Response: For the inter-comparison experiment, participants were provided with the RGI6 outline of each study site and requested to provide elevation change estimates within that area. Therefore, we did not compare the results with other glacier inventories like GAMDAM Glacier Inventory as it was beyond the scope of the experiment, as reported in line 762: "In addition, there are some issues that were not covered in the present inter-comparison experiment, but that can have an important impact on geodetic glacier mass change assessments."

We used RGI6, as stated in line 148 because at the time of the experiment, RGI7 had not yet been released. We also received estimates from the Hugonnet et al. (2021) study that relied on RGI6 within the experiment.

Line No. 775: Density conversions: It was unclear what value of density was considered for the study 900-850? Clarify.

Response: As for the glacier area changes, the issue of density conversion is not part of this experiment. However, in the discussion section 5.2, we listed and described all the factors that can affect the elevation change estimates, including the problem associated with the density conversion (line 775). So, here, we do not refer to any specific values as this section is a broad description of the issue; instead, we reported the two most commonly used values in the community for the density conversion.
To clarify, the density conversion factor of 850 kg m−3 following Huss (2013) was used in this experiment for the temporal correction of the estimates provided by the participants, using the methods developed by Zemp and Welty (2023) as reported in section 2.6 and the discussion (line 728).

---

## Author Comment (AC2)

Title: **Observing glacier elevation changes from spaceborne optical and radar sensors – an inter-comparison experiment using ASTER and TanDEM-X data**

Author(s): Livia Piermattei et al.
MS No.: egusphere-2023-2309
MS type: Research article

Handling editor: Vishnu Nandan

**Author's response related to the comments received to the manuscript:** https://doi.org/10.5194/egusphere-2023-2309

We thank the two reviewers and the editor for their feedback on our manuscript. We revised our manuscript under consideration of their comments and suggestions, which helped to improve the clarity of our paper.

The main changes we made to the manuscript are as follows:
- We improved the method and discussion sections by clarifying the challenges for optical and radar data and the solutions implemented by the participants within their processing chain.
- Throughout the manuscript, we improved the description and clarity of the challenges of radar data, especially in mountainous areas, and the impact of radar penetration on the assessment of elevation changes.
- We adjusted Figure 4 by adding a list of all participants for each group.

We are convinced that these revisions - as detailed in the point-by-point reply below - addressed all the comments raised by reviewer 1 and the editor (based on the feedback from reviewer 2) and made our manuscript more understandable.

**Point-by-point reply to reviewer comments**: editor and reviewer comments are in black, and the authors' response in blue, with citations from the revised manuscript in green.

……………………………………………………………………….

**RC1**: 'Comment on egusphere-2023-2309', Silvan Leinss, 16 Jan 2024

Review of "Observing glacier elevation changes from spaceborne optical and radar sensors – an inter-comparison experiment using ASTER and TanDEM-X data" by Livia Piermattei et al. EGUsphere-2023-2309.

**General comment**:

The paper is well written and almost completely clear to understand. It is well structured and contains all relevant information. The intercomparison of glacier DEM differences identifies the most critical and important processing steps and also indicates which processing steps are less relevant to the results. The work is very valuable to judge on the reliability of optical or radar DEM differences for glacier height changes. I have only a few minor specific comments, mainly addressing radar-specific topics for DEM generation over glaciers.

Response: Thank you for taking the time to review our manuscript and provide us with very valuable feedback. We greatly appreciate your constructive comments and interest in our work. We have addressed the minor comments you provided, and we included the implementation in the updated version of our manuscript as reported here below in green.

The authors note the lack of time-series based approaches to estimate glacier elevation change in their study; below I provide several relevant insights that can be learned from analyzing radar DEM time series over glaciers based on an IEEE publication from 2021, including an openly available dataset with 140 TanDEM-X DEMs from Aletsch Glacier (2011-2019).

Response: Thank you for pointing out this interesting study on the TanDEM-X time series on the Aletsch Glacier. While we cannot integrate these results into our experiment, we will cite them in the manuscript and discussion sections and compare the findings with our results on the Aletsch Glacier.

**Specific comments:**

250: "as reference for surface elevation for TanDEM-X production": Do you mean "for production of the TanDEM-X DEMs (as described in section 2.5.2)"? - To avoid confusion with the TanDEM-X DEM from DLR/Airbus or the TanDEM-X mission.

Response: We agree this needs clarification. For the production of our TanDEM-X DEMs (as described in section 2.5.2), we apply differential interferometry using the SRTM DEM as a reference. The processing of our raw DEMs follows the approach by Sommer et al. (2020) without the final co-registration or mosaicking, as this was the task for the intercomparison exercise. We have clarified this in the text.
Line 252: "The SRTM DEM is used as reference surface elevation for TanDEM-X DEM production based on differential interferometry following the approach by Sommer et al. (2020) (as described in section 2.5.2). Additionally, the SRTM DEM is directly used to estimate the elevation change rates of the Baltoro and Northern Patagonian Icefield for the period 2000−2010s."

Sommer, C., Malz, P., Seehaus, T.C. et al. Rapid glacier retreat and downwasting throughout the European Alps in the early 21st century. Nat Commun 11, 3209 (2020). https://doi.org/10.1038/s41467-020-16818-0

336: "... a mean or median ... is calculated, starting with the DEM with the smallest date difference to the validation": I don't understand this sentence. a mean or median is calculated over a set of data; why would a mean/median start with some specific DEM pair?

Response: Thank you, and we agree that the sentence needs to be clarified.
Line 344: "A similar approach named "DEM mosaic" groups the DEMs in a time window of approximately two years around each target date. Then, for each DEM group, the mean or median of all overlapping cells is computed to estimate the elevation value and corresponding acquisition date. Another DEM mosaic solution uses the DEM closest to the target period first and then the remaining voids are filled with the other DEMs selected within the time window based on their acquisition date. Consequently, the elevation value and its respective acquisition date extracted from the DEM groups may vary from cell to cell."

343: "vertical deformation resulting from sensor behaviour...": I hope, the used sensors do not cause any deformation of the observed glaciers :) I guess, you mean offsets or spatially varying biases in the produced DEMs as written in the next sentence. I think you can simply drop "vertical deformation" here.

Response: We agree that it was not correctly formulated. We rephrased the sentence as follows.

Line 355: "Sensor behaviour, data acquisition, and processing can systematically bias relatively large fractions of the DEMs, resulting in incorrectly measured spatial trends and vertical deformation…."

346: "ramps/constant offsets/phase jumps": I doubt that ramps are caused by phase-jumps, at least not by locally constrained phase-jumps that appear as a well visible constant offset over a bounded area. Furthermore, ramps indicate linear (or higher order) trends rather than "constant offsets". I guess, ramps could also be caused by small (cm-scale) errors in the given satellite state vectors. The TanDEM-X orbits have at least centimeter-accuracy, but an error of 1.5 cm line-of-slight would already correspond to a full phase cycle in the interferogram and a driftig error would cause a phase ramp. Therefore, I think that phase ramps could originate from orbit inaccuracies. See also (Krieger 2007, page 3331-3332). Of course, phase jumps would cause other kinds of errors and could also lead to ramps when fitting a plane over an area with locally bounded phase-unwrapping errors.
Reference: G. Krieger et al., "TanDEM-X: A satellite formation for high-resolution SAR interferometry," IEEE Trans. Geosci. Remote Sens., vol. 45, no. 11, pp. 3317–3341, Nov. 2007, doi: https://doi.org/10.1109/TGRS.2007.900693.

Response: Thank you for pointing out this wrong formulation. We rephrased the sentence, clarifying that ramps are not caused by phase jumps.
Line 359: "TanDEM-X formation baseline errors cause tilts in range. Additionally, radar scenes can be affected by spurious elevation jumps originating from phase unwrapping errors ("phase jump"), which have a magnitude of multiples of the height of ambiguity (Rizzoli et al., 2017). Furthermore, phase ramps might also partially originate from the SRTM reference DEM due to along-track undulations with a frequency of several kilometres (Farr et al., 2007)."

Rizzoli et al. Generation and performance assessment of the global TanDEM-X digital elevation model, ISPRS Journal of Photogrammetry and Remote Sensing, Volume 132, 2017, Pages 119-139, ISSN 0924-2716, https://doi.org/10.1016/j.isprsjprs.2017.08.008.

348: It might be irrelevant here, but the SRTM also shows along-track undulations with a frequency several kilometers that are caused by oscillations of the boom where the second radar antenna was attached. If the SRTM was involved as any auxiliary data for DEM generation, mentioning these oscillations here might be relevant.
T. G. Farr et al., "The Shuttle Radar Topography Mission," Rev. Geophys., vol. 45, pp. 1–33, 2007, doi: 10.1029/2005RG000183. (section 5.3) and G. Franceschetti, A. Iodice, S. Maddaluno, and D. Riccio, "Effect of antenna mast motion on X-SAR/SRTM performance," IEEE Transactions on Geoscience and Remote Sensing, vol. 38, no. 5, pp. 2361–2372, 2000, doi: 10.1109/36.868892.

Response: Thank you for your comment. We included the potential phase ramp error introduced by SRTM in the text by citing the suggested reference. Please see the full sentence in our reply to the previous comment. (Line 362).

375: "bias...ignored if they were smaller than the pixel resolution": Could you specify this a bit more? I guess, the coregistration approach contains already a resampling of the DEM even when the shifts were well below a pixel resolution. Otherwise, an offset of 1 pixel (assuming 30 m pixel spacing) could cause a height error of 30m on 45° inclined terrain. What exactly is meant with "remaining biases"? I guess, all remaining biases are in the z-direction because the figure-of-merit function is usually the height-error between two DEMs? Or are there any other remaining biases in the horizontal direction?

Response: We agree that this can be better clarified. The resampling depends on the co-registration algorithm; here, we are referring to the remaining vertical bias as, ideally, there is no remaining horizontal bias after co-registration. To correct the remaining vertical bias, one could add or subtract the related offset. Solutions adopted by the participants included applying a correction for the across and along-track shifts in ASTER DEMs and removing a first-order spatial polynomial vertical bias. We clarified this in section 3.3.2 Spatial bias correction. So, we removed this sentence in the co-registration section (Line 393) to avoid confusion and repetitions.

Line 370: "... DEM generation as proposed by (Girod et al., 2017). After co-registration, additional vertical bias corrections were applied to the ASTER DEMs. UGA removed a first-order spatial polynomial vertical bias, while LEG corrected the across and along-track shifts based on Gardelle et al. (2013)."

381ff: Could you also mention artifacts due to wet snow or terrain with very low backscatter? When the backscatter drops below the noise-equivalent-zero, the coherence is completely lost. I guess, the impact is similar to radar shadow.

Response: Thank you. We agree, and the mentioned effects are added to the manuscript as follows.

Line 400: "...Artefacts in the TanDEM-X DEM are produced by phase unwrapping errors and low or lost coherence in areas with low backscatter or even below the noise-equivalent-zero. This problem especially occurs in mountainous regions, where wet snow areas have very low backscatter values. Additionally, steep slopes and vertical cliffs cause artefacts and voids in side-looking InSAR acquisition geometry due to shadowing and layover."

402: For penetration biases, you could add a reference to Leinss (2021), Section VI-D: S. Leinss and P. Bernhard, "TanDEM-X: Deriving InSAR height changes and velocity dynamics of Great Aletsch Glacier," IEEE J. Sel. Topics Appl. Earth Observ. Remote Sens., vol. 14, pp. 1–18, 2021, doi: 10.1109/JSTARS.2021.3078084.

Response: Thank you for the suggestion. We added the reference in the manuscript as follows.

Line 423: "... but average penetration depths of up to several metres (> 5 m) have been reported previously on Aletsch Glacier (Leinss et al., 2021; Bannwart et al., 2023) and other mountain glaciers in the Alps and High mountain Asia (Dehecq et al., 2016; Li et al., 2021) as well as the ice sheets (Abdullahi et al., 2018; Rott et al., 2021), while for the longer wavelength C-band of SRTM, …"

612-613: I suggest to provide numbers objectively describing the deviations from the validation dataset rather than the subjective words "perfectly fit", "underestimate", "overestimate" because order of magnitude of "perfectly fit" and "under/overestimate" in Fig. 11 is almost the same. - although within the standard error (also here: provide a number). Could you mention if the means of ASTER and TanDEM-X include or exclude the low-confidence results?

Response: Thank you for this comment. We improved the text by adding the reference to the figure legend in the text, which was missing, and we also reported that the ensemble means are calculated excluding low-confidence results. This information was only stated in the figure caption. However, we prefer not to add the values here as the corresponding mean values and standard error are in the figure legend.

Line 645: "...The ensemble mean of the full experiment sample (excluding low-confidence results) perfectly fits the validation result, while the ensemble means of the ASTER and TanDEM-X underestimate and overestimate, respectively, the validation data, although within

the standard error of the corresponding sample. Values are reported in the legend of Figure 11."

624: "large spread between the experiments": Could you provide a number how larger the spread is (between results with reasonable confidence)?

Response: We agree. We reported the range of the spread in m per year, excluding low confidence results, for the three study sites and referred to the supplement table for all dh results. Line 657: "...that the large spread between the experiment results (ranging from 0.7 to 1.0 meter per year, as shown in Tables S16-S18 under "dh_T2_final") represents a major challenge…"

633: "none of the groups used a time series approach with the TanDEM-X data, partly, to avoid issues with radar penetration, which is expected to have a larger impact in cold winter than in wet summer snow conditions":

Certainly, radar penetration is a major uncertainty for radar interferometry-based DEMs, however, Leinss and Bernhard (2021) used a time series of ~140 TanDEM-X acquisitions over Aletsch glacier. In Section VI-C (p.4808) the authors observe that their height time series (Fig. 11) show the smallest variability between December and January, indicating relatively constant penetration during this period. In consequence they used only the winter(!) acquisitions for temporal regression of their time series, because during summer and especially during spring, the observed height changes vary significantly stronger due to variable penetration (dry/wet snow) and ice melt in summer. For the period 2011-2019 they observed a height change rate of -1.54 m/yr for the observed area (59 km^2). The observed area was reduced compared to the 78 km^2 given in Fig 1 (this paper) due to radar layover and shadow affecting mainly the accumulation area where height changes were negative, but relatively small (layover affected to a minor fraction also the glacier tongue with significant height changes of -3m/yr). A back-of-the envelope calculation, referencing to the 78 km^2 (Fig. 1, this paper), results in a height change rate of (-1.54 m/yr * 59 km^2 + 0.3 m/y * (78 km^2 - 59 km^2))/78 km^2 = 1.24 m/yr which is close to the validation value of −1.14 ± 0.15 m/yr (Table 1, this paper). The TanDEM-X DEM time series are available online: https://doi.org/10.3929/ethz-b-000482456

Reference: Leinss and P. Bernhard, "TanDEM-X: Deriving InSAR height changes and velocity dynamics of Great Aletsch Glacier," IEEE J. Sel. Topics Appl. Earth Observ. Remote Sens., vol. 14, pp. 1–18, 2021, doi: 10.1109/JSTARS.2021.3078084.

Response: Thank you for clearly describing the results obtained using a time series approach with TanDEM-X and providing the radar penetration findings. We also appreciate the comparison of this study with our inter-comparison results. We added this reference and your key insights regarding radar penetration (see green text below). However, the observation periods and areas were different and, hence, not entirely comparable, and hence, we decided not to report these dh/dt values in the discussion.

Line 666: "We note that none of the groups used a time series approach with TanDEM-X data. This could be due to the experimental setup, with much fewer available DEMs than those used, for example, by Leinss and Bernhard (2021), who extracted the elevation change of Aletsch glacier using a time series of approximately 140 TanDEM-X acquisitions. A dense DEM time series is required to evaluate periods of stable signal penetration throughout the year. Leinss and Bernhard (2021) observed that their elevation change time series shows the smallest variability between December and January, indicating relatively constant penetration during this period. As a result, they used only these winter acquisitions for temporal regression of their time series because, during summer and especially spring, the observed elevation changes vary significantly due to variable penetration caused by dry/wet snow and ice melt in summer."

713-735: Section "Differences in Survey Period": I think this section is mainly valid for optical data and not well suited for radar data because height changes due to penetration can be stronger than height changes due to accumulation. As discussed in Section VI-C in Leinss (2021), reference above, the apparent (radar-)height decreases in early winter (Oct-Dec) because refreezing of firn causes more penetration bias that the additional accumulation bias due to fresh snow. Furthermore (adressing line 731), radar data ist least suitable for comparison of spring DEMs because penetration varies most strongly during spring.

Response: Thank you for this comment. We agree that for TanDEM-X, the temporal correction based on the glaciological method is not really suited. This is why TanDEM-X results were submitted already temporally scaled by most groups. We clarified this point in the manuscript as follows, and we added the finding about early winter penetration into the radar penetration subsection.
Line 758: "…no correction as the DEM selection was already close to target dates, corrections filling temporal gaps with annual glaciological observations, and corrections based on long-term annual trends derived from the selected DEMs. While the first two strategies mainly applied to optical DEMs, they were not well-suited for radar data. Most groups used winter radar DEMs, and the few tests with TanDEM-X DEM from the end of the ablation season were considered low confidence due to unrealistically low values of elevation change over ten years (e.g., Hintereis DLR-1 and UIO-5, Fig. 7 and Table S16). Therefore, TanDEM-X results were submitted already temporally scaled by most groups. Long-term trends were scaled by…."

Line 802: "This bias, which can be as high as 8 meters, aligns with the findings of recent studies by Leinss and Bernhard (2021) and Bannwart et al. (2023) conducted in the same location, while time series analysis for other regions even revealed higher seasonal signals (Vijay and Braun, 2017). According to Leinss and Bernhard (2021), the height changes due to penetration changes can be stronger than height changes due to accumulation, and the apparent (radar-)height decreases in early winter (Oct-Dec) because refreezing of firn causes more penetration bias than the additional accumulation bias due to fresh snow."

745: "radar penetration of a few decimeter per year": Here, an absolute dh value should be given, rather than a change rate. Penetration matters less for larger time intervals between DEMs.

Response: We agree, thank you for this point. We added the absolute values and clarified the sentence.
Line 791: "In our experiments, only two groups – UST and LFR – addressed this issue by applying radar penetration corrections for Baltoro, while others included this in their uncertainty estimate. The corrections made by UST and LFR were a few decimetres per year, or 1.2 m and 2.1 m, respectively, over 2000-2012 when using SRTM and almost negligible in the case of two TanDEM-X DEMs (LFR group), with a correction of 0.07 m per year over the period between 2012 and 2019."

Caption, Figure 14: A summer winter comparison is not only biased by penetration but equally by ice melt especially at lower elevations. The text mentions this, but it needs also to be mentioned in the caption of the figure.

Response: We agree, thank you for the suggestion. We updated the figure caption as follows.
Line 811: "Figure 14: The potential impact of radar penetration at high elevations and the ice melt at lower elevations on the Aletsch glacier when two TanDEM-X DEMs from different seasons are subtracted. The elevation change of summer (2013.08.11) and winter (2013.03.21)

DEMs are shown with the glacier hypsometry (grey bars, second y-axis). For elevation bands with limited areas (less than 0.1 km$^2$), the elevation change is shown as an empty circle."

770: "in the glacier forefield": add: "and at the glacier margins", because less ice thickness usually comes along with a reduced ice extend at all glacier margins.

Response: We agree, thank you. We added it to the manuscript.
Line 808: "...or no ice loss in the glacier forefield and at the glacier margins (within RGI outlines)."

678, 719, 806 vs. 630: The selection of the best acqusition time seems to differ between optical and radar data. While for optical data, the end of the ablation season seems to be the best option, for radar data DEM pairs from the mid-winter seems to be best suited as the radar penetration ist most similar in deep winter (line 630, and Leinss 2021). In the accumulation area (and only there), radar height changes appear still significant at the end of the ablation season, probably due to refreezing of firn (Fig. 11(a) in Leinss 2021).

Response: Thank you for this comment. We agree that a distinction between optical and radar scene selection strategies is appropriate. However, if for Aletsch and similar regions, the same mid-winter-to-winter penetration has been observed, we believe this cannot be generalised, as this strongly depends on the regional setting (e.g., how much surface melt is there, is it across the entire glacier, and so on ...) and might be completely different for other regions. We rephrased this section accordingly.
Line 859: "Optimal selection of DEMs based on their quality and timing is essential for accurately detecting long-term trends while minimising noise and seasonal variability. However, the optimal acquisition time for optical and radar data seems to differ. For optical data, the best option is to select DEM pairs of good quality at the annual mass-balance minimum, which occurs at the end of the ablation season. This helps to avoid biases due to snow accumulation and related temporal corrections. Conversely, for radar data, the mid-winter period appears to be the most suitable time for DEM pair acquisition since radar penetration is most similar in deep winter (Leinss and Bernhard, 2021). In the accumulation area, radar height changes still appear significant at the end of the ablation season, probably due to the refreezing within the firn (Fig. 11a, Leinss and Bernhard, 2021), but this is dependent on the regional setting (e.g., distribution and location of surface melt). Mosaic or time series approaches can help to reduce the effects of random error associated with any individual DEM (optical or radar) and extract additional information from the DEM stacks."

851: "steep terrain during phase unwrapping": I would split that in two parts: Radar missions have to cope with steep terrain (causing terrain masking by radar shadow and layover) and with phase unwrapping (most difficult in steep terrain due to masking and a high fringe-frequency).

Response: Thank you for the suggestion. Also, we would like to add that with more up-to-date and accurate reference DEMs, the phase unwrapping process can be supported, and fewer phase unwrapping errors can be expected. Shadow and layover can't be avoided except with multiple acquisition geometries.
Although the suggestions are quite detailed for a conclusion section, we agree with this separation and implement the suggestions in the updated manuscript.
Line 910: "The experiments also revealed clear challenges with each data source. Optical imagery has some limitations in notoriously cloudy or textureless regions (the latter holding especially for sensors with limited dynamic range), which can result in gaps and noise in the DEM. The quality of DEMs from radar missions can deteriorate in steep terrain due to the masking effect caused by radar shadow and layover. Due to masking and a high fringe

frequency, phase unwrapping is also challenging in such areas. Additionally, radar data can be affected by wet snow, which has very low backscatter and can cause correlation loss, as well as seasonal and annual variable radar penetration."

851: Possibly relevant is also wet snow with very low backscatter causing correlation loss.
Response: Thank you, this is true, and it is an additional argument for selecting DEM scenes mid-winter instead of the end of ablation in this region. We added it to the manuscript, see the sentence above.

**Technical corrections/typos:**

306: remove by before reference or use format "by Hugonnet et al. (2021)"
Response: Thank you, implemented.

344: remove "at a scale at a horizontal scale" -> "at a horizontal scale
Response: Thank you for spotting, removed.

---

## Author Response (AR2)

Title: **Observing glacier elevation changes from spaceborne optical and radar sensors – an inter-comparison experiment using ASTER and TanDEM-X data**

Author(s): Livia Piermattei et al.
MS No.: egusphere-2023-2309
MS type: Research article

Handling editor: Vishnu Nandan

**Author's response related to the comments received to the manuscript:**
https://doi.org/10.5194/egusphere-2023-2309

We thank Silvan Leinss for providing additional clarification regarding the radar data, specifically regarding the optimal selection of radar DEMs for assessing glacier elevation changes. We have implemented all the suggestions, which has enhanced the clarity of the paper with respect to radar data.

In addition, we checked all our figures to ensure they were colourblind-friendly by selecting colours from the colorbrewer2 tool (https://colorbrewer2.org/). Please note that for some figures, we had to use 12 different colours, and there is no palette available that can fulfil all colour vision deficiencies, as suggested in the Color Blindness Simulator. Nevertheless, the content of the figure can also be read even without colours, such as the flow directions (Fig. 4) and the total number of values for each class reported in the figure (e.g., Fig. 8).

**Point-by-point reply to reviewer comments**: editor and reviewer comments are in black, and the authors' response in blue, with citations from the revised manuscript in green.

………………………………………………………………………….

**RC1**: 'Comment on egusphere-2023-2309', Silvan Leinss, 14 May 2024

Review of "Observing glacier elevation changes from spaceborne optical and radar sensors – an inter-comparison experiment using ASTER and TanDEM-X data" by Livia Piermattei et al. EGUsphere-2023-2309.

**Specific/technical comments**:

l. 132: "Furthermore, high-relief topography can (...) cause geometric DEM distortions in radar images": What do you mean here exactly? any 3D relief causes distortion when imaged in 2D (optical or radar), but I guess, that's not what you mean here. Why would the relief distort the DEM?

Response: Here, we generically describe the challenges of the different study sites for optical and radar data. We meant that radar imagery is susceptible to geometric distortions in high mountain areas due to foreshortening, layover, and shadow and in areas affected by the layover and shadow, phase unwrapping is difficult, resulting in inconsistencies in the DEM.
We correct the sentence by removing the reference to "DEM distortion" and clarify the issue of radar data in high-relief topography.
"Furthermore, high-relief topography can cast shadows in the optical images and cause geometric DEM distortions in radar images due to foreshortening and layover effects."

Section 3.3.1 and Figure 5: approach vs. differencing strategy: The terms "self-processed", "DEM pair" (Fig. 5: "Pair"), "DEM mosaic" (Fig. 5 "Mosaic"), "DEM time series" (Fig. 5. "Time series") are sometimes called "strategy", sometimes "approach", sometimes the approaches include "DEM" sometimes not (Fig. 5). To make the manuscript easier to read, try to use consistent terms and call them all approach OR strategy. The different approaches could be described in individual subsubsubsections or, maybe preferably, in a list so that the reader can quickly find the referenced approach.

Response: Agreed. We corrected the text and figure 5 and consistently used the term "approach" throughout the paper. We prefer not to add a list of the different approaches, and we clarified that a detailed description of the adopted approach is provided in the supplementary table of each group.

l. 352: "tilts in range": Replace "in range" by "in the DEM" and add reference (Krieger, 2007). Even though the tilt appears mainly in across-track direction (ground range), along-track variations of the baseline errors result in along-track tilts. Both tilts sum up to some tilt of the DEM.

Response: Done. Thank you for the clarification.

l. 388 check sentence: "(...) in areas with low backscatter or even below the noise-equivalent-zero." -> "in areas with backscatter below the noise-equivalent-zero". End of line: "in in" -> "in"

Response: Done.

l. 391: "artifacts can be detected and removed": remove "and removed" or replace with a more suitable term. e.g. "filled", "corrected", ...

Response: Done.

l. 408: "large for dry and cold snow and ice": remove "and ice". Because for blue ice, the apparent penetration depth is almost zero because almost all scattering occurs at the air/ice interface - even though there is significant energy transfer into the ice volume (but this energy is not scattered back but absorbed in the ice).

Response: Done. Thank you for the explanation.

l. 408 "and increased moisture content": you could even say: "penetration depths decrease significantly/drastically with increasing moisture content".

Response: Done.

l. 512: "The TanDEM-X results show different signals in the large accumulation area, with several results indicating more negative elevation changes(...)": it is not clear what "different signals" exactly means. It seems you could shorten this sentence to "The TanDEM-X results indicate more negative elevation changes in the accumulation area ..."

Response: Done. We applied the suggestion by specifying that "Most of the TanDEM-X results indicate..."

l 702: "acquired at the end of the glacier ablation period provides maximum control during the processing workflow." - This seems to be the best choice for ASTER, but, according to the discussion l.647-648, for "radar data, winter DEMs were preferred". Please add this information here.

Response: Agreed and added it in the text.
"As such, a pair approach with two high-quality DEMs, with few data voids, acquired at the end of the glacier ablation period for optical data, or winter DEMs for radar, provides maximum control during the processing workflow."

l. 758: "In general, we can conclude that the selection of DEMs close to the annual mass-balance minima (around the 1st of October for northern mid-latitudes) is a good strategy to reduce the impact of differences in survey periods.": I think this is best for optical data (ASTER), but as discussed in l. 653-658, for radar (TanDEM-X), selection of DEMs up to 2-3 months after the mass-balance minimum (Dec - Jan for northern mid-latitudes) results in elevations close or even slightly below the elevation at the anual mass-balance minimum because of penetration through the fresh winter snow and even into the refreezing firn and thereby provides the smallest impact when differences in survey periods need to be corrected.

Response: Thank you. Agreed and implemented.
"In general, we can conclude that selecting optical DEMs close to the annual mass-balance minima (around the 1st of October for northern mid-latitudes) is a good strategy to minimise the impact of differences in survey periods. For radar (TanDEM-X), we recommend selecting DEMs after the mass-balance minima, up to 2-3 months later (i.e., Dec - Jan for northern mid-latitudes) when elevations are close to or slightly below the elevation at the annual mass-balance minima due to penetration through the fresh winter snow and even into the refreezing firn."

l. 760 "can be an option": Please differentiate also here: This is only an option for optical data. Radar DEMs show the highest variability (due to snow melt) and uncertainty (due to low backscatter) close to the annual mass-balance maxima.

Response: Agreed and implemented.
"DEM selection close to the annual mass-balance maxima (between March and May) can be an option only for optical data, as radar DEMs show the highest variability (due to snow melt) and uncertainty (due to low backscatter). However, even with optical DEMs, variability is high depending on the accumulation history, and it is more subject to material density uncertainty associated with the elevation change to glacier mass change conversion. In any case, selecting the same season for start and end dates is important to avoid bias in the derived trend."